# Allosteric modulation of peroxisomal membrane protein recognition by farnesylation of the peroxisomal import receptor PEX19

Leonidas Emmanouilidis[1,2,*], Ulrike Schütz[1,2,*], Konstantinos Tripsianes[3,*], Tobias Madl[1,2,4], Juliane Radke[5], Robert Rucktäschel[5], Matthias Wilmanns[6], Wolfgang Schliebs[5], Ralf Erdmann[5] & Michael Sattler[1,2]

The transport of peroxisomal membrane proteins (PMPs) requires the soluble PEX19 protein as chaperone and import receptor. Recognition of cargo PMPs by the C-terminal domain (CTD) of PEX19 is required for peroxisome biogenesis *in vivo*. Farnesylation at a C-terminal CaaX motif in PEX19 enhances the PMP interaction, but the underlying molecular mechanisms are unknown. Here, we report the NMR-derived structure of the farnesylated human PEX19 CTD, which reveals that the farnesyl moiety is buried in an internal hydrophobic cavity. This induces substantial conformational changes that allosterically reshape the PEX19 surface to form two hydrophobic pockets for the recognition of conserved aromatic/aliphatic side chains in PMPs. Mutations of PEX19 residues that either mediate farnesyl contacts or are directly involved in PMP recognition abolish cargo binding and cannot complement a ΔPEX19 phenotype in human Zellweger patient fibroblasts. Our results demonstrate an allosteric mechanism for the modulation of protein function by farnesylation.

[1] Institute of Structural Biology, Helmholtz Zentrum München, Ingolstädter Landstr. 1, Neuherberg 85764, Germany. [2] Munich Center for Integrated Protein Science at Chair of Biomolecular NMR, Department Chemie, Technische Universität München, Lichtenbergstr. 4, Garching 85747, Germany. [3] CEITEC—Central European Institute of Technology, Masaryk University, Kamenice 5, Brno 62500, Czech Republic. [4] Institute of Molecular Biology and Biochemisty, Medical University of Graz, Graz 8010, Austria. [5] Institute of Biochemistry and Pathobiochemistry, Department of Systems Biology, Faculty of Medicine, Ruhr University Bochum, Bochum 44780, Germany. [6] EMBL Hamburg, Notkestr. 85, Geb. 25A, 22607 Hamburg, Germany. * These authors contributed equally to this work. Correspondence and requests for materials should be addressed to K.T. (email: kostas.tripsianes@ceitec.muni.cz) or to M.S. (email: sattler@helmholtz-muenchen.de).

Peroxisomes are ubiquitous organelles present in all eukaryotic cells with pivotal roles in cellular homoeostasis. They catalyse reactions in lipid metabolism and decompose hydrogen peroxide, as well as numerous other toxic compounds. The biological importance of peroxisomes is highlighted by a number of inherited diseases associated with malfunctions of peroxisomal proteins[1]. Single-enzyme defects like acatalasia can give rise to comparably mild phenotypes, whereas defects in peroxisomal biogenesis, for example, in the Zellweger syndrome, are lethal. Peroxisomal proteins are directed to the organelle posttranslationally via distinct transport systems, which are specific for either matrix or membrane proteins[2]. The molecular mechanisms of matrix protein recognition for transport into peroxisomes are well characterized. Cycling receptors recognize the matrix proteins in the cytosol, direct them to a docking complex at the peroxisomal membrane, where import of the folded proteins takes places through a transient import pore[3,4]. In comparison, our knowledge on the transport of peroxisomal membrane proteins (PMPs) is still scarce. Peroxisomal biogenesis factor 19 (PEX19) undoubtedly is a key player in several steps of PMP transport. First, it is thought to function as a chaperone for newly synthesized PMPs in the cytosol[5]. Second, PEX19 directs the cargo to the peroxisomal membrane, where it docks to the transmembrane protein PEX3 thereby acting as a shuttling receptor[6]. Third, it could be involved in membrane insertion of PMPs (refs 7,8). Finally, the transfer of PEX3 from the ER to the peroxisome, which is an early step in peroxisome biogenesis, occurs in a PEX19-dependent manner[9].

PEX19 is posttranslationally modified by farnesylation. In spite of an overall low sequence similarity across PEX19 homologues, the farnesylation site is conserved throughout evolution with an exception of trypanosomal PEX19 (ref. 10). The farnesyl group is a C15 isoprenoid (Fig. 1a), which is covalently attached to target proteins by farnesyltransferase[11]. This enzyme catalyses the attachment of the farnesyl group from farnesyl pyrophosphate (substrate) to a cysteine residue of a C-terminal signal sequence called CaaX box ('C' denotes the modified Cys, 'a' an aliphatic amino acid and 'X' usually stands for Ser, Thr, Gln, Ala or Met). Farnesylation and geranylgeranylation, a C20 isoprenoid modification, are classified as prenylation and are irreversible post-translational modifications[12] mostly found in small GTPases, that is, Ras and Rho proteins[13]. Human PEX19 is farnesylated in vivo[14,15]. It was reported that PEX19 is completely farnesylated in Saccharomyces cerevisiae and that the affinity for PMP cargo peptides is ten-fold increased with farnesylated PEX19 (ref. 16). Furthermore, farnesylation-deficient yeast exhibits reduced stability of PMPs and suffers from defects in peroxisomal biogenesis in vivo[16]. It has been suggested that farnesylation may alter the conformation of PEX19, but the molecular and functional consequences of PEX19 farnesylation are so far unknown.

Human PEX19 consists of 299 amino acids with an intrinsically disordered N-terminal half that interacts with the membrane-bound docking protein PEX3 (refs 6,17,18) and PEX14 (ref. 19). The folded C-terminal domain (CTD) mediates binding to PMPs and harbours the CaaX box site for farnesylation[20] (Fig. 1a). A crystal structure of a C-terminal fragment (comprising residues 161–283)[21], referred to as CTDΔC below, showed that the PEX19 fold comprises four α-helices that exhibit a three-helical bundle domain and an additional N-terminal helix α1 protruding away from this bundle. In vitro binding studies demonstrated that the CTDΔC fragment is capable of binding PMP peptides with micromolar affinity, while mutational analysis suggested that residues located in helix α1 contribute to cargo binding[21]. However, the crystallized PEX19 fragment lacks the C-terminal 16 amino acids, which include the farnesylation site, and thus the structural impact of farnesylation and its role in the modulation of PMP binding remain unknown.

Here, we present the solution structure of the farnesylated C-terminal PMP binding domain of human PEX19 and report molecular details for the recognition of hydrophobic residues in PMPs determined by NMR spectroscopy. NMR data indicate that the C-terminal residues of the CTD become rigid upon farnesylation. Surprisingly, the PEX19 CTD undergoes significant conformational changes to accommodate the farnesyl group inside a large hydrophobic cavity, which, in turn, affect functional interactions of PEX19 with PMPs. On the basis of NMR chemical shift perturbations and intermolecular NOEs we identify two hydrophobic binding pockets for aromatic residues in PMPs, which are formed upon farnesylation of PEX19. Mutations interfering with either farnesyl recognition or PMP interactions affect the PMP binding in vitro and the biological activity of PEX19 during peroxisome biogenesis in vivo. Our data suggest that farnesylation orchestrates the PMP binding region of PEX19 by an allosteric mechanism. Thereby, farnesylation contributes PMP binding and the functional activity of PEX19 in PMP import. These findings reveal a novel mechanism for the modulation of protein function by farnesylation.

## Results

**Structure of the farnesylated PEX19 CTD**. To investigate the effects of farnesylation, we studied the human PEX19 CTD (residues 161–299) that includes the native C-terminus with the CaaX box of the protein (Fig. 1a). PEX19 CTD is structurally autonomous and does not interact with the preceding disordered region of the full-length protein as judged by NMR fingerprint spectra (Supplementary Fig. 1a). Recombinant farnesyltransferase was used for farnesylation of PEX19 CTD in vitro. An increased electrophoretic mobility in SDS–polyacrylamide gel electrophoresis (SDS–PAGE) (Supplementary Fig. 1b) and NMR spectra (Fig. 1b) demonstrate virtually complete farnesylation of the PEX19 CTD. NMR $^{13}C^{\alpha/\beta}$ secondary chemical shifts indicate that the four α-helices observed in the crystal structure of the truncated PEX19 CTDΔC (residues 161–283) are also present in the farnesylated protein (Supplementary Fig. 1c,d). However, farnesylation induces large differences in amide chemical shifts throughout the PEX19 CTD that involve many residues remote from the farnesylation site (Cys296) (Fig. 1b,c). Low $\{^1H\}$-$^{15}N$ heteronuclear NOE values and large solvent paramagnetic relaxation enhancements (solvent PREs) for the C-terminal ten residues in the non-farnesylated PEX19 CTD indicate that they are highly flexible in solution and solvent exposed, respectively. In contrast, a significant increase of heteronuclear NOE, and reduced sPRE values in the farnesylated PEX19 CTD (Fig. 1c) demonstrate that farnesylation strongly reduces the backbone flexibility and solvent accessibility of the C-terminal residues. Similar sPRE changes, although to a lesser extent, are also observed for residues in the N-terminal helix α1 (Fig. 1c). These data demonstrate that farnesylation induces a substantial rigidification and compaction of the PEX19 CTD that primarily stabilizes the C-terminal region but also locks the arrangement of helix α1 with respect to the core domain. Comparable tumbling correlation times ($\tau_c$) derived from $^{15}N$ NMR relaxation data for non-farnesylated ($\tau_c = 11.4$ ns) and farnesylated ($\tau_c = 10.5$ ns) PEX19 CTD demonstrate that both protein conformations are monomeric in solution (Supplementary Fig. 1c,d).

To unravel the structural changes induced by farnesylation we determined the structure of the farnesylated PEX19 CTD (Fig. 2a–c). Given the presence of substantial signal overlap, a combination of optimized isotope-labelling with isotope-filtered NOESY experiments was acquired to obtain unambiguous

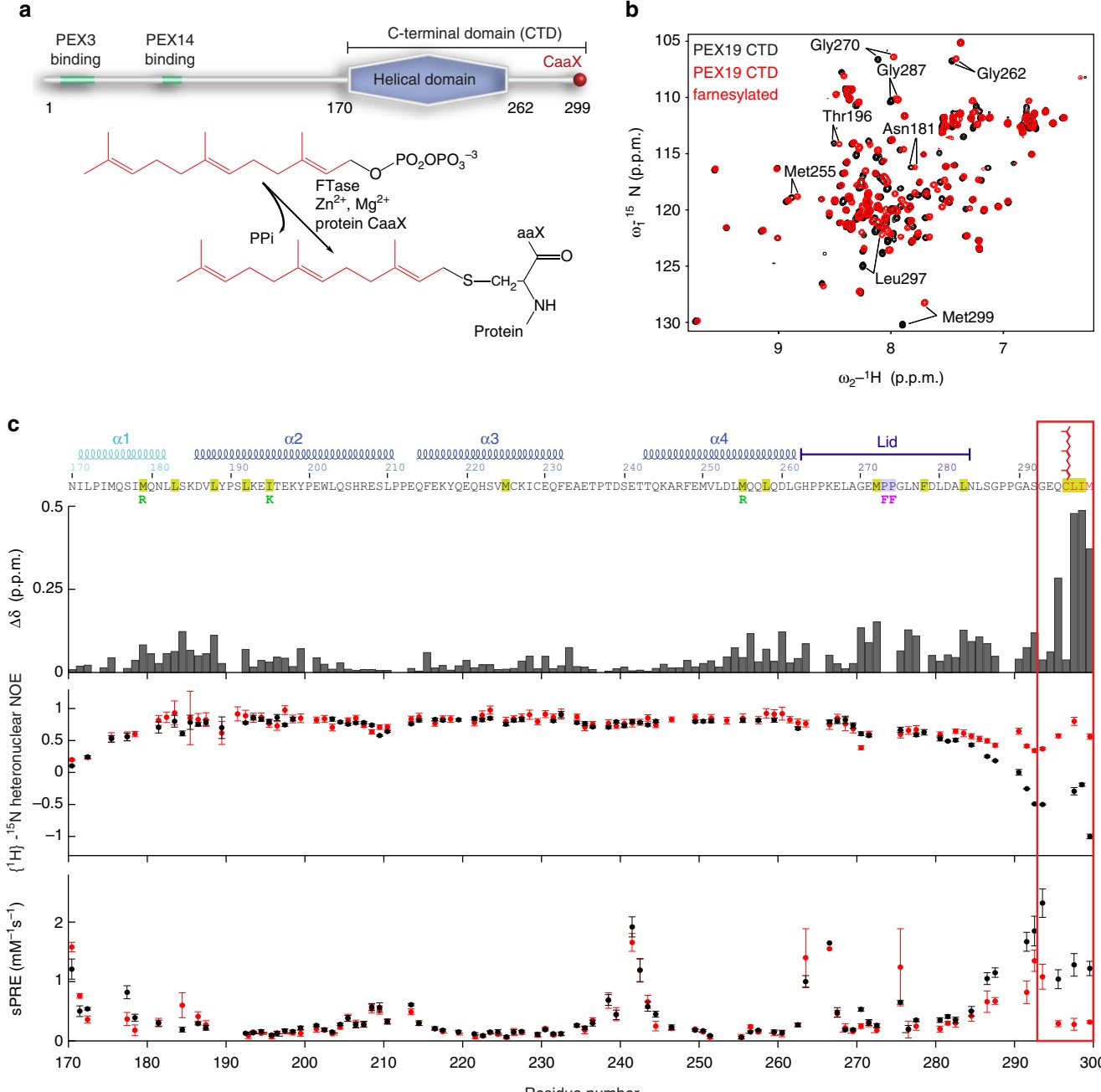

**Figure 1 | NMR analysis of PEX19 farnesylation.** (**a**) Schematic overview of human PEX19: the unfolded N-terminus of PEX19 interacts with the peroxisomal membrane proteins PEX3 and PEX14, the C-terminus harbours the α-helical cargo binding region and the farnesyl recognition sequence CaaX. Farnesyl transferase catalyses the transfer of the farnesyl moiety from farnesyl pyrophosphate to the cysteine of the CaaX box. (**b**) Overlay of $^1$H,$^{15}$N HSQC spectra of PEX19 CTD with (red) and without (black) farnesylation. Amide resonances that are strongly affected by farnesylation are annotated. (**c**) Chemical shift perturbation (Δδ) of amide protons induced by farnesylation (top), {$^1$H}-$^{15}$N heteronuclear NOE (middle) and solvent paramagnetic relaxation enhancement (sPRE) rates for amide protons (bottom) are shown for PEX19 CTD with (red) and without (black) farnesylation. Error bars for heteronuclear NOE data based on s.d. of noise in the individual spectra were calculated using error propagation. Error bars for sPRE data represent fitting error of proton $R_1$ relaxation rates recorded from different concentrations of Gd(DTPA-BMA). The large changes in the C-terminal region that harbours the CaaX box are highlighted by a red box. The amino acid sequence and secondary structure elements of the PEX19 CTD are indicated on top, with aliphatic residues involved in farnesyl binding highlighted in yellow. Point mutations of residues that contact the farnesyl group or involved in PMP recognition are indicated as green and purple letters, respectively.

assignments of chemical shifts and NOEs (Supplementary Fig. 2a). The solution structure of the farnesylated PEX19 CTD was determined based on numerous NOE-derived distance restraints including 203 protein-farnesyl restraints and 136 residual dipolar couplings (Fig. 2; Supplementary Fig. 2b;

Table 1) and was further validated by comparison of experimental and back-calculated solvent PREs for amide protons (Supplementary Fig. 2c). The structure of the farnesylated PEX19 CTD contains the N-terminal helix α1 followed by the three-helical bundle domain (helices α2, α3 and α4) and a linker that

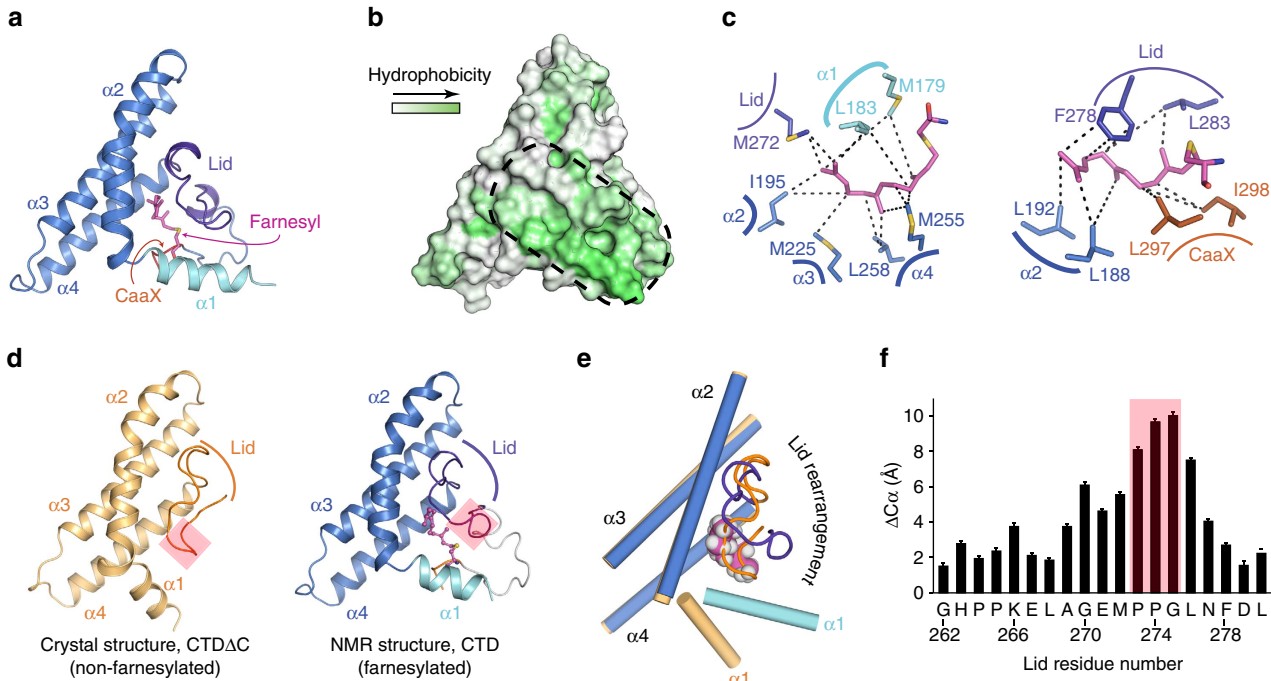

**Figure 2 | Structure of farnesylated PEX19 CTD.** (**a**) Cartoon representation of farnesylated PEX19 CTD. The farnesyl binding site is formed by residues in helix α1 (cyan), helices α2–α4 (blue), the lid region (residues 262–283; violet) and the CaaX box (orange). The farnesyl is drawn as sticks in magenta. (**b**) Surface representation of the farnesylated PEX19 CTD coloured by hydrophobicity. The hydrophobic surface region comprising helix α1 and the lid is indicated by a black dotted line. (**c**) Two different views of the farnesyl recognition site. Dashed lines indicate intermolecular NOE correlations, collected on PEX19 farnesylated samples using various labelling schemes on the protein side. The relative position of helices in the PEX19 CTD fold surrounding the farnesyl is indicated. (**d**) Comparison of the crystal structure of the PEX19 CTD lacking the C-terminal 16 residues (CTDΔC, left)[21], with the solution structure of the farnesylated PEX19 CTD (right). Structural changes of the lid region are highlighted by a red box. (**e**) The structure of the core helical bundle comprising helices α2–α4 is highly similar while the orientation of helix α1 (cyan) differs. The lid (residues 262–283) covers the farnesyl group in farnesylated PEX19 CTD (violet), while in the crystal structure of non-farnesylated PEX19 CTDΔC these residues occupy the farnesyl binding cavity (orange). Note, that residues 283–299 in the NMR structure and loop regions connecting the helices have been omitted for clarity. (**f**) Large conformational changes of the lid induced by PEX19 farnesylation are indicated by change in the Cα atom positions of lid residues with the helical core domain superimposed as in **e**. Error bars indicate the s.d. for the 20 models of the NMR ensemble. The largest changes are highlighted by a red box as in **d**.

connects α4 and the CaaX box (Fig. 2a). All four helices form a hydrophobic cavity that contacts the farnesyl group with aliphatic side chains: Met179, Leu183 in helix α1; Leu188, Leu192, Ile195 in helix α2; Met225 in helix α3; Met255, Leu258 in helix α4 (Fig. 2c). Residues within the linker region between α4 and the CaaX box (Met272, Phe278 and Leu283) mediate additional hydrophobic interactions to the farnesyl moiety and cover the binding cavity like a lid (Fig. 2a,c). The two aliphatic residues of the CaaX box (Leu297 and Ile298) contact the first isoprene unit and complete the binding pocket thus largely shielding the farnesyl group from the solvent. The farnesyl group is occluded in this internal cavity with a buried surface area of 457 Å² as determined by PDBePISA (ref. 22). The small size of the farnesyl binding cavity in the PEX19 CTD may thus stabilize a distinct bent conformation of the isoprenoid (Fig. 2; Supplementary Fig. 3).

The fold of the core three-helical bundle of farnesylated PEX19 CTD (helices α2, α3 and α4) is similar to the recently reported crystal structure of non-farnesylated PEX19 CTDΔC (ref. 21) (residues 161–283) (Fig. 2d) with a backbone coordinate r.m.s.d. of 0.84 ± 0.06 Å (residues 187–261). However, a number of important structural elements rearrange in a unique manner to form a continuous hydrophobic cavity and bury the farnesyl group, thus shielding it from the solvent. In the non-farnesylated protein the lid segment associates with other parts of the structure to shield hydrophobic patches and essentially occupies the position of the lipid (Fig. 2d,e). When compared to the non-farnesylated crystal structure of PEX19 CTDΔC, the most prominent differences involve a specific orientation of helix α1 to accommodate the bulky farnesyl group (Fig. 2e), and the displacement of the lid (Fig. 2f). Both elements form hydrophobic interactions with the farnesyl group and adopt a unique rigid arrangement relative to the three-helical bundle with the tip of the lid packing against the region between helices α1 and α2 (Fig. 2d). In the crystal structure of the non-farnesylated protein helix α1 is involved in a non-physiological tetramerization of the PEX19 CTDΔC (ref. 21), which defines its orientation. In contrast, the non-farnesylated PEX19 CTD (residues 161–299) is monomeric in solution (Supplementary Fig. 1c). Moreover, NMR relaxation data (Supplementary Fig. 1d) and the scarcity of intramolecular NOEs indicate that helix α1 is partially flexible and does not adopt a specific arrangement relative to the core helical fold. Thus, the structure of the farnesylated protein and our NMR data for the non-farnesylated protein demonstrate that farnesylation stabilizes a specific orientation of helix α1 and reshapes the lid region. Most importantly, the recognition of the farnesyl group in the internal cavity induces allosteric changes at the protein surface to form a hydrophobic groove (Fig. 2b) that serves as the binding site to PMP cargo peptides (see below).

**Mutational analysis of farnesyl recognition.** On the basis of the structure of farnesylated PEX19 CTD, we mutated residues that form the farnesyl binding pocket (M179R in helix α1, I195K in

**Table 1 | NMR and refinement statistics for farnesylated PEX19.**

| | |
|---|---|
| *NMR restraints* | |
| Distance restraints | |
| Total no. of NOEs | 3,728 |
| Intra-residue | 823 |
| Inter-residue | 2905 |
| Sequential ($|i-j|=1$) | 1,100 |
| Medium-range ($|i-j|<4$) | 1,002 |
| Long-range ($|i-j|>5$) | 600 |
| Protein-farnesyl | 203 |
| Total no. of dihedral angle restraints | 187 |
| $\phi$ | 91 |
| $\psi$ | 96 |
| Total no. of RDCs | 136 |
| $Q^{RDC}$ (%) | 28 |
| | |
| *Structure statistics* | |
| Violations (mean ± s.d.) | |
| Distance restraints (Å) | 0.018 ± 0.00 |
| Dihedral angle restraints (°) | 0.913 ± 0.06 |
| Max. dihedral angle violation (°) | 4.75 |
| Max. distance restraint violation (Å) | 0.46 |
| Deviations from idealized geometry | |
| Bond lengths (Å) | 0.003 ± 0.00 |
| Bond angles (°) | 0.511 ± 0.07 |
| Impropers (°) | 1.175 ± 0.04 |
| Ramachandran plot statistics (%) | |
| Residues in most favored regions | 91.5 |
| Residues in additionally allowed regions | 8.2 |
| Residues in generously allowed regions | 0.2 |
| Residues in disallowed regions | 0.1 |
| Average pairwise r.m.s.d. (Å)* | |
| Backbone | 0.45 ± 0.09 |
| Heavy atoms | 0.85 ± 0.11 |

*Pairwise coordinate r.m.s.d. was calculated for 20 refined structures for residues 175–285 and 295–299.

helix α2 and M255R in helix α4, Fig. 2c; Supplementary Table 1) to impair hydrophobic contacts and assess the functional importance of specific protein–lipid interactions. [1]H, [15]N NMR correlation spectra (Supplementary Fig. 4a) of the corresponding PEX19 CTD variants show that all mutant proteins studied maintain their structural integrity. Although these protein mutants are completely farnesylated *in vitro*, the introduction of the farnesyl group leads to differential chemical shift changes compared to the wild-type protein. Chemical shift differences of M179R and I195K versus the wild-type PEX19 CTD in the non-farnesylated and farnesylated state mainly involve residues adjacent to the mutation site, consistent with local effects introduced by replacing the corresponding residues (Supplementary Fig. 4b). In contrast, pronounced and wide-spread effects are seen for the M255R mutant, suggesting that the recognition and conformation of the farnesyl group in the PEX19 M255R variant is altered. To probe whether the M255R mutation leads to partial exclusion of the farnesyl moiety from the binding cavity, we compared the changes in {[1]H}-[15]N heteronuclear NOE values for the wild type and M255R PEX19 CTD upon farnesylation. Notably, while similar values are observed for the helical core domain, the farnesylation-induced increase of heteronuclear NOE values for the C-terminal residues seen for wild type PEX19 is less pronounced in the M255R PEX19 variant (Supplementary Fig. 4c). The reduced NOE values indicate an increased flexibility of the C-terminal region. The fact that the tail does not become completely flexible as seen for the non-farnesylated protein implies that the farnesyl group is not completely excluded from the cavity in the M255R PEX19 CTD variant. In any case, the significantly

increased flexibility of the C-terminal region is consistent with a weakened protein-farnesyl interaction and suggests a partial and/or transient exposure of the hydrophobic farnesyl group in the M255R variant.

To provide further evidence for the (partial) solvent exposure of the farnesyl group in the mutant proteins, we employed hydrophobic interaction chromatography. Differences in surface hydrophobicity were determined for farnesylated and non-farnesylated wild type and mutant PEX19 CTD proteins (Fig. 3a; Supplementary Fig. 4d). As expected, the introduction of the farnesyl group generally increases the hydrophobicity of PEX19 CTD for both wild type and mutants. However, the change in hydrophobicity linked to farnesylation is significantly larger for all PEX19 CTD variants and increases from M179R to I195K to M255R. This is consistent with an increased exposure of the farnesyl group in these mutants, presumably due to steric clashes and/or charges introduced in the binding cavity. The hydrophobic interaction chromatography analysis shows the most pronounced effects for the M255R substitution consistent with a significant exposure of the farnesyl moiety in the M255R variant (Fig. 3a).

We next examined the role of farnesylation for the functional activity of PEX19. We transfected PEX19-deficient cells with a bicistronic vector expressing PEX19 variants and eGFP-SKL as peroxisomal targeting signal 1 (PTS1) marker. The PEX19-deficient fibroblasts are characterized by the absence of import-competent peroxisomes, which is indicated by the mislocalization of the peroxisomal matrix marker to the cytosol and mislocalization of the peroxisomal membrane marker PEX14 to mitochondria[23]. Complementation and thus reappearance of import-competent peroxisomes is indicated by an overlapping punctate pattern for eGFP-SKL and PEX14. Full-length wild type and PEX19 variants with mutations that affect the farnesyl interactions studied above were compared in a complementation assay in human *ΔPEX19* fibroblast cells. Whereas, wild type PEX19 restores peroxisomal protein transport to 60%, functional complementation was reduced to 46% in PEX19 C296S and further decreased to 23% for PEX19 I195K. The M179R and M255R mutations are not capable of rescuing the *ΔPEX19* phenotype (Fig. 3b,c). As shown by immunoblot analysis, wild type and mutant proteins are expressed in similar amounts and are farnesylated *in vivo*—with the exception of a C296S variant, which lacks the farnesylation site (Fig. 3d). Statistical analysis of the complementation efficiency reveals that peroxisomal protein import is partially impaired in the absence of farnesylation (C296S). Mutants that affect the recognition of the farnesyl group and the PMP cargo binding (I195K, M179R) strongly impair peroxisome formation. Notably, no rescue of peroxisome biogenesis is observed in a PEX19 mutant (M255R) that interferes with intramolecular farnesyl recognition, even though this residue is remote from the PMP binding surface. We also considered the possibility that the complementation defects of the PEX19 CTD mutants were caused by altered localization due to the exposure of the farnesyl moiety resulting in increased hydrophobicity. To this end, immunofluorescence microscopy revealed that import defects do not correlate with stronger association of PEX19 with peroxisomes or mislocalization to any other membrane-bound subcellular compartment (Supplementary Fig. 5).

**Recognition of PMPs by farnesylated PEX19 CTD.** Residues in helices α1 and α2 (residues 170–195) are highly conserved and have been previously implicated in binding of PMP cargo proteins[21]. Our structural analysis shows that farnesylation leads to a conformational rearrangement of this region and the lid in

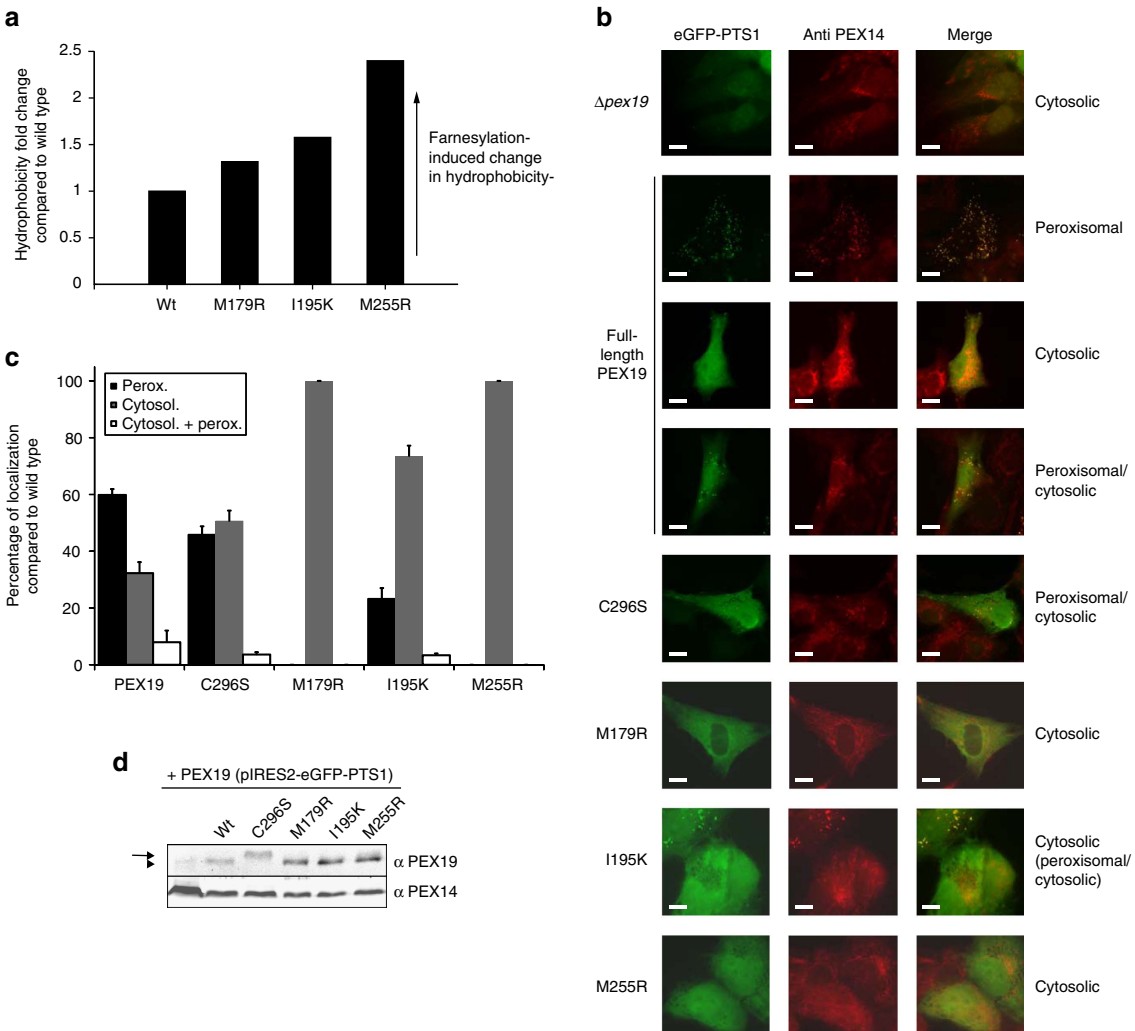

**Figure 3 | Functional and biochemical analysis of PEX19 variants that affect farnesyl recognition.** (**a**) Hydrophobic interaction chromatography analysis. Proteins were eluted from a Butyl Sepharose FF column using a linear gradient of decreasing $(NH_4)_2SO_4$ concentrations. For PEX19 CTD wild type and every variant, the farnesylated protein exhibits an increased elution volume compared to the non-farnesylated form (Supplementary Fig. 4d). The plot shows relative differences in hydrophobicity of the farnesylated and non-farnesylated state for each protein, compared to the wild type protein (which was set to 1). (**b**) Functional complementation of PEX19-deficient fibroblasts by PEX19 variants. PEX19 harbouring single-amino acid substitutions as indicated were introduced into PEX19-deficient fibroblasts by transfection with bicistronic expression vectors coding for full-length PEX19 variants and eGFP-PTS1. Localization of the model peroxisomal protein transport substrate eGFP-PTS1 was monitored by fluorescence microscopy (left column). Endogenous PEX14 as a peroxisomal membrane protein was detected using immunofluorescence microscopy (middle column). A punctate staining pattern and co-localization of eGFP-PTS1 and PEX14 indicate functional peroxisomal protein transport (merge, right column). For better comparison, PEX19-deficient cells expressing only eGFP-PTS1 from the vector are shown in the upper panels. Scale bar: 10 µm. The designation indicated on the right refers to the GFP-localization. (**c**) Quantitative analysis of $\Delta PEX19$-phenotype complementation by the individual PEX19 variants. Values are obtained from analysing at least 100 cells in three independent transfection experiments. Data shown represent mean ± s.d. (**d**) Immunoblot analysis of the farnesylation mutants shows that all variants are expressed at wild type level and, except for C296S (arrow), are farnesylated *in vivo* (arrowhead).

PEX19 CTD. To map the binding surface with PMPs, we monitored NMR chemical shift perturbations upon addition of peptides that comprise previously characterized PEX19 binding regions[24] to [15]N-labelled farnesylated or non-farnesylated PEX19 CTD (Fig. 4). Probably, due to the hydrophobicity of the PMP-derived peptides their addition to PEX19 often resulted in precipitation. However, for a peptide derived from the ATP-binding cassette (ABC) transporter ALDP we could obtain soluble complexes at various PEX19-peptide ratios and analyse NMR chemical shift perturbations. Upon addition of the ALDP peptide to non-farnesylated PEX19 CTD, no significant spectral changes are observed at molar ratio up to 10:1 (Fig. 4a). In contrast, addition of the same peptide to farnesylated PEX19

CTD induces distinct chemical shift perturbations (Fig. 4b,c). The amide and methyl signals affected by the titration cluster to the hydrophobic groove that is formed on the surface of the farnesylated PEX19 structure (Figs 2b; 4d,e). The affinity for the ALDP peptide was measured by fluorescence polarization (Supplementary Fig. 6). For the wild type protein, farnesylation increased the affinity seven-fold, in agreement with findings reported for yeast PEX19 (ref. 16) and consistent with the NMR titration data. In contrast, all mutants (M179R, I195K, M255R) showed impaired binding with or without farnesylation (Supplementary Fig. 6).

We also compared the binding of other PMP-derived peptides to farnesylated and non-farnesylated PEX19 CTD using NMR

and microscale thermophoresis experiments. To this end, binding of a peptide derived from the PMP PEX13 induces even stronger NMR chemical shift perturbation changes to helix α1, and significant line broadening in the lid region, highlighting the importance of the lid in PMP binding (Supplementary Fig. 7a,b). For a PEX11-derived peptide we could quantitatively compare changes in binding affinity induced by farnesylation. These data show, that the interaction with non-farnesylated PEX19 CTD is reduced to a $K_D = 29.8 \pm 1.6\,\mu M$ compared to $K_D = 6.1 \pm 0.4\,\mu M$ for the farnesylated protein (Supplementary Fig. 7c).

The molecular recognition of the ALDP PMP peptide and farnesylated PEX19 CTD is documented by chemical shift perturbations and a large number of intermolecular NOEs observed mainly between methyl groups supporting further the methyl chemical shift perturbations (Fig. 4; Supplementary Fig. 8a, see methods for details). Few intermolecular NOEs could be assigned for the PEX19–PMP interaction involving methylene or methyl groups of PEX19 and the aromatic phenylalanine sidechain of the ALDP peptide. These intermolecular NOEs, chemical shift perturbations, and previously published mutational analysis[21] were used as ambiguous interaction restraints to calculate a structural model of the farnesylated PEX19 CTD–PMP peptide complex using HADDOCK (ref. 25). As the complex is not sufficiently stable to obtain assignments of the PEX19-bound form, the helical conformation of the docked peptide was inferred from secondary chemical shifts of the free peptide, which indicate a preformed helical conformation (Supplementary Fig. 8b). In fact most PMPs exhibit hydrophobic or aromatic residues spaced by three residues that are exposed on the same face of an α-helical conformation (Fig. 5a).

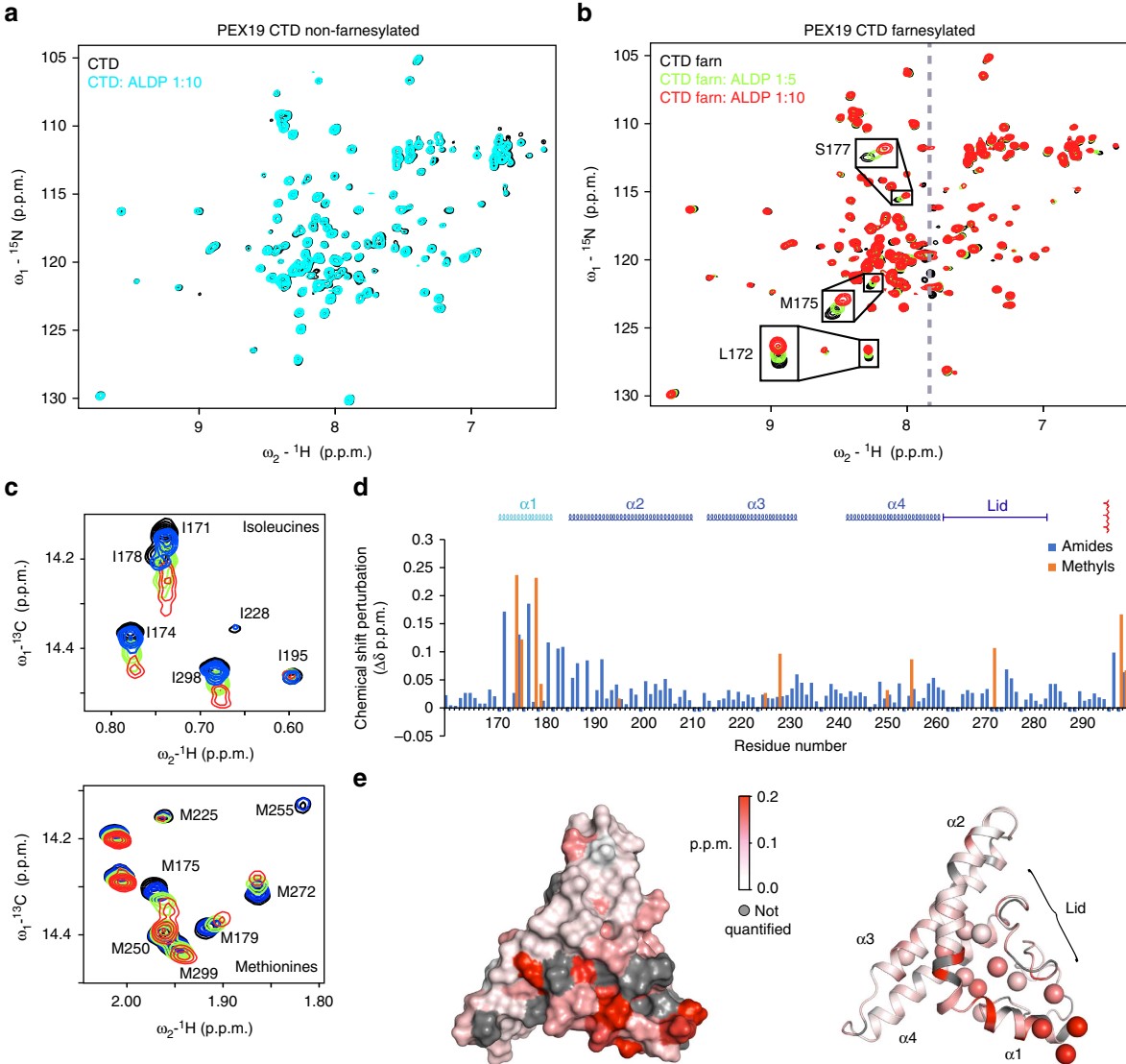

**Figure 4 | NMR analysis of PMP binding to non-farnesylated and farnesylated PEX19 CTD.** (**a**) Comparison of ¹H,¹⁵N HSQC spectra of 100 μM non-farnesylated CTD free (black) and in the presence of 10-fold excess of the ALDP-derived peptide (cyan). (**b**) Comparison of ¹H,¹⁵N HSQC spectra of 100 μM farnesylated PEX19 CTD free (black) and in the presence of 5 (green) and 10 (red) molar ratio of the ALDP peptide. The grey dashed line indicates the position of additional baseline correction applied to remove noise from DMF signals in the peptide stock solution. (**c**) Chemical shift perturbation of Met/Ile methyl groups seen in ¹H,¹³C HSQC experiments upon increasing concentration of ALDP peptide. Black: free PEX19 CTD 100 μM, blue, green and red correspond to 200, 500 and 1,000 μM ALDP peptide concentration, respectively. (**d**) Quantification of chemical shift perturbation (CSP) of amide and methyl groups with respect to the PEX19 amino acid sequence. (**e**) Left: Surface representation of the PEX19 CTD coloured in red according to CSP. Right: cartoon representation with the backbone coloured based on amide CSP in red and spheres depicting methyl CSPs upon addition of the ALDP peptide (right). Grey colour refers to residues, which could not be analysed.

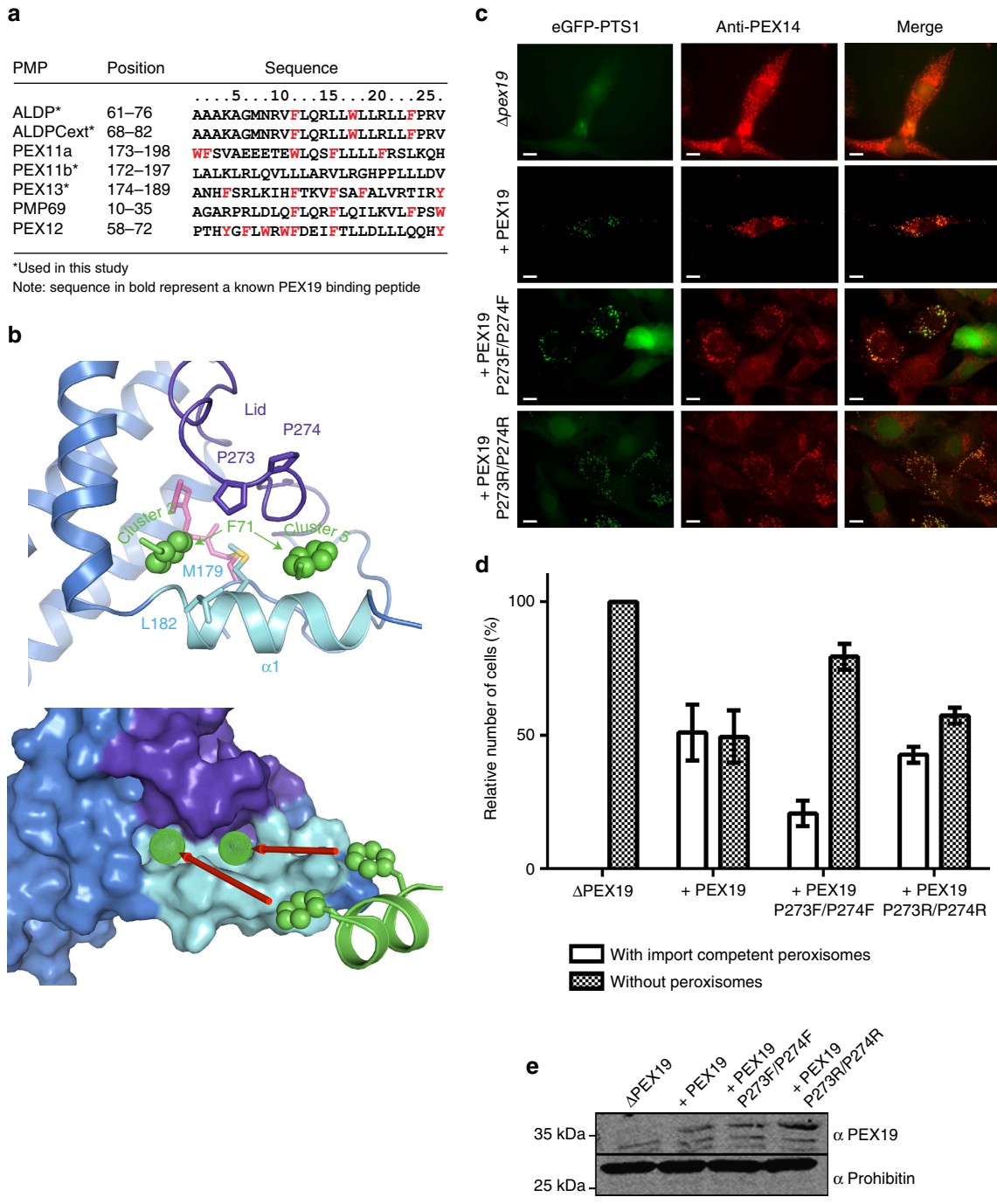

**Figure 5 | Recognition of a PMP-derived peptide by the farnesylated PEX19 CTD.** (**a**) Amino acid sequences of peroxisomal membrane proteins. Aromatic residues are highlighted in red; regions known to be involved in PEX19 binding are shown in bold. (**b**) Docking models showing two binding pockets for aromatic side chains of the PMP-derived peptide (green stick and ball representation). Top: The PEX19 lid and helix α1 are coloured dark blue and cyan, respectively. Side chains of M179, Leu182, Pro272 and Pro273 are shown as sticks. Bottom: Surface representation of the PEX19 CTD indicating the hydrophobic cavities that accommodate the aromatic side chain of Phe71 in the ALDP peptide. (**c**) In vivo effects of mutations of the PMP binding site of PEX19. Immunofluorescence microscopy images of PEX19-deficient human fibroblasts transfected with bicistronic vectors encoding for eGFP-PTS1 and different PEX19 variants. The same plasmid lacking PEX19 was used as a negative control (ΔPEX19) showing diffuse staining due to mitochondrial mislocalization for both peroxisomal marker proteins, eGFP-PTS1 (matrix protein) and PEX14 (PMP). The same plasmid lacking PEX19 was used as a negative control (ΔPEX19) showing cytosolic and mitochondrial mislocalization for the peroxisomal marker proteins eGFP-PTS1 (matrix protein) and PEX14 (PMP), respectively. A congruent punctate pattern of eGFP-SKL and PEX14 indicates formation of import-competent peroxisomes 72 h after transfection. Scale bar: 10 µm. (**d**) Statistical analysis of number of transfected cells containing import-competent peroxisomes. Values were obtained from 100 cells each of three independent transfected experiments. Data shown represent mean ± s.d. (**e**) Immunoblot analysis of mutants shows that all variants are present in an amount similar to the wild type protein.

The docking calculations yielded two clusters of structures, which differ with respect to the orientation of the ALDP peptide and the protein (Fig. 5b; Supplementary Table 2). This is explained by the scarcity of intermolecular restraints, which provide only distance information for the phenylalanine ring, and thus cannot unambiguously define the orientation of the PMP helix. Nonetheless, in both clusters an aromatic side chain occupies either of two hydrophobic pockets that are formed by the rearrangement of helix α1 and lid segment upon farnesylation (Figs 2e and 5b). This is consistent with previous mutational analysis[21] where disruption of helix α1 abolishes PMP binding by PEX19. However, our docking model also indicates that residues in the lid, that is, two proline residues (Pro273, Pro274), are involved in PMP recognition (Fig. 5b). Notably, the size and hydrophobic character of the PMP binding surface suffice to accommodate two hydrophobic residues of a PMP (Fig. 5b).

The physiological contribution of residues in the PMP binding site was probed by mutational analyses in vivo. Previous studies have already implicated residues in helix α1 in PMP binding in vitro and for peroxisome function in cells[21]. However, the role of residues in the lid has not been explored. Therefore, we studied the contributions of Pro273 and Pro274, two residues in the PEX19 lid that are implicated in ALDP/PMP binding (Fig. 5b; Supplementary Table 1) by substitution with arginine or phenylalanine. The phenylalanine substitutions were designed with the hypothesis that they would occupy the binding pockets for aromatic residues in the PMP cargo proteins and thus interfere with binding in cis. Full-length wild type and variant PEX19 proteins were expressed in PEX19-deficient human fibroblasts, which are characterized by the lack of peroxisomes. Cells that are not complemented by functional PEX19 show a cytosolic mislocalization of the peroxisomal marker protein eGFP-PTS1 and a diffuse background immunolabelling of the PMP PEX14, which is distinct from the peroxisomal pattern and known to reflect mislocalization to mitochondria (Fig. 5c, ΔPEX19). After 72 h transfection, 50% of cells expressing wild type PEX19 display formation of newly import-competent peroxisomes as indicated by the congruent punctate pattern of both marker proteins (Fig. 5b, +PEX19). Cells, which express the PEX19 P273F/P274F mutant, displayed a significantly lower complementation rate of only 20% (Fig. 5d), independent of the PEX19 protein level (Fig. 5e). The impaired function of the PEX19 mutant demonstrates that the lid region is not only important for binding of the half-transporter ALDP but also for other PMPs, in particular for those which are essential for biogenesis of peroxisomes.

## Discussion

Our structural analysis shows that the farnesyl group in the PEX19 CTD is buried within a hydrophobic cavity. Note, that this cavity has been found to be empty in the previously reported crystal structure[21]. Consistent with this, substantial conformational differences are seen in the farnesylated PEX19 CTD when compared to the crystal structure of the non-farnesylated PEX19 CTDΔC (ref. 21). An unexpected finding of or structural analysis is that PEX19 farnesylation stabilizes a distinct conformation of helix α1 and the C-terminal region of PEX19 and thereby arranges the PMP binding site, suggesting an allosteric modulation of the PMP interaction. Farnesylation affects PMP binding by preorganization of the PMP binding surface with an appropriate placement of the helix α1 and the lid and thereby provides a hydrophobic binding surface for PMP binding. The functional significance of this allosteric control of PMP binding is confirmed by our mutational analysis in vitro and in vivo, which shows that PEX19 mutations that affect farnesyl

recognition remote from the PMP binding site strongly interfere with peroxisome function.

To our knowledge, the structure of the farnesylated human PEX19 CTD is the first example of a protein, which buries the farnesyl group by an internal cavity formed by the same polypeptide chain in cis. A recent structural analysis of a farnesylated Ras protein found that the farnesyl group is recognized by the guanine nucleotide dissociation inhibitor (GDI) PDEδ in trans and is released in an allosteric manner by the G proteins Arl2/Arl3 (refs 26,27). Such an allosteric control is reminiscent of the modulation of PMP binding by PEX19 by farnesylation in cis as observed in the present study.

Other structurally well-characterized examples of isoprenoid binding are farnesyltransferases (Supplementary Fig. 3). In these enzymes, conserved aromatic amino acids interact with the isoprene units of the substrate, thereby recognizing the farnesyl in an extended conformation[28]. Farnesyl recognition by PEX19 is distinct from farnesyl transferase complexes, as mainly aliphatic residues contact the farnesyl, which adopts a bent conformation. The bent conformation of the farnesyl group is presumably linked to the small size of the PEX19 fold (Supplementary Fig. 3).

Recent studies have shown that non-farnesylated PEX19 recognizes PMP cargo proteins less efficiently. Accordingly, S. cerevisiae cells lacking the farnesylation motif are deficient in growth on oleic acid as sole carbon source, a metabolic condition that depends on functional peroxisomes[16]. Here, we show that farnesylation of the PEX19 CTD leads to an enhanced PMP binding. We note that additional effects of PEX19 farnesylation could contribute to the functional activity of the PEX19 protein. (1) The presence of a farnesyl group may influence the PMP membrane insertion, that is, farnesyl exposure could support PMP cargo release by interactions with the membrane, PEX3 or other factors. (2) It is also important to note that farnesylation may be only a first step of posttranslational modification of PEX19. It is conceivable that the farnesylated CaaX box is further processed by C-terminal proteolysis and carboxymethylation. This modification could be a way to regulate the activity of PEX19, as the carboxymethylation will further enhance the hydrophobicity of the C-terminal region and thus modulate interactions at the membrane. The presence of this PEX19 modification in cells has not yet been experimentally determined and its functional role should be investigated in future studies.

The structural analysis shows that a specific conformation of the conserved PMP binding site is assembled and controlled by farnesylation of the PEX19 CTD. Our NMR data and docking calculations demonstrate that the PMP binding maps to the region comprising helix α1 and the lid region[21], involving substantial structural rearrangements (Fig. 2e,f). The groove formed on the surface by the lid and helix α1 is sufficiently large to accommodate hydrophobic side chains that are spaced by about three amino acids, that is, corresponding to a single helical turn in the PMP ligand peptides. This is fully consistent with the hydrophobic nature of PMP peptides and an α-helical conformation that—as observed for the ALDP peptide—is partially preformed, already in the absence of PEX19. The observation of two cavities, which can accommodate aromatic and/or aliphatic side chains exposed from a helical peptide suggests optimal features of membrane protein peroxisomal targeting signals (mPTS) in peroxisomal membrane proteins. Indeed all PMPs exhibit hydrophobic mPTS motifs, which are predicted to adopt an α-helical conformation and comprise aromatic or aliphatic side chains in a spacing of 3–4 residues.

For the recognition of PMPs, helix α1 can be thought of as the thumb and the lid as the fingers of a hand, which, upon farnesylation, grasps the PMP (Fig. 6). Previously identified mutations on helix α1, which probably disrupt the helix, have

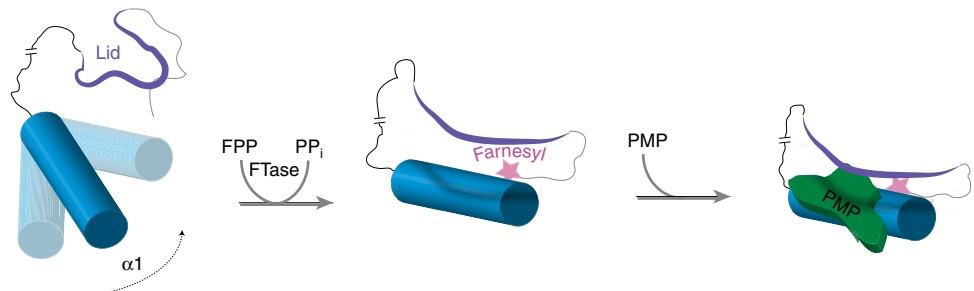

**Figure 6 | Schematic model for the allosteric regulation of PMP binding by farnesylation.** In case of non-farnesylated PEX19 CTD, the helix α1 is flexible in solution and only loosely attached to the helical core domain of the PEX19 CTD, while the lid adopts an open conformation. Farnesylation induces a specific, rigid arrangement of these two structural elements, thus forming a hydrophobic cavity, which then forms a high-affinity binding site for PMPs.

been shown to strongly affect the PMP binding[21]. Here, we provide direct structural evidence that both helix α1 and the lid, which is stabilized in a unique conformation by farnesylation of the PEX19 CTD, provide key interactions for PMP binding. In particular, replacement of two consecutive prolines in the lid by phenylalanine interferes with PMP binding and therefore affects peroxisomal biogenesis.

Our mutational analysis shows that mutation of M255, which is important for farnesyl recognition but remote from the PMP binding site, significantly impairs the PMP interaction (Fig. 3). This strongly argues that farnesylation stabilizes a specific conformation of the binding surface and thereby enhances the binding affinity for PMP cargo, as we have determined for different PMP-derived peptides in the present study. The structural rearrangements coupled to PEX19 farnesylation therefore suggest an allosteric control of PMP binding by farnesylation (Fig. 5).

To investigate the functional importance of the farnesyl recognition by PEX19, we monitored the functional activity of mutant proteins that impair farnesyl binding. The mutations introduced may influence PMP binding in two ways.

Mutations may affect direct contacts with the PMP cargo and thereby reduce functional activity. Schueller et al have previously shown that mutations of conserved hydrophobic residues in helix α1, such as Ile178 and Leu182, or truncation of this helix abolish PMP binding and the functional activity of PEX19 (ref. 21). Consistent with these observations, the M179R variant in helix α1 investigated in the present study cannot restore peroxisomal biogenesis in ΔPEX19 fibroblasts. This mutation presumably destabilizes the orientation of helix α1 and consequently disturbs the formation of the hydrophobic pocket (see NMR chemical shift perturbations, Fig. 5b; Supplementary Fig. 4b). Here, we demonstrate that the binding site also involves the lid, as proline residues in the lid have an equally important contribution to the binding (Figs 5b and 6).

On the other hand, mutations may impair functional activity by affecting PMP binding indirectly via allosteric conformational changes induced at the PMP binding surface. The role of farnesylation is demonstrated by the PEX19 C296S variant, which is not farnesylated and cannot fully restore PEX19 function. The functional activity of the PEX19 I195K variant is also impaired although less strongly, presumably as Ile195 contacts the distal part of the farnesyl group and thus has smaller contributions to farnesyl recognition. In contrast, the M255R variant completely abolishes PEX19 function in fibroblasts, even though the mutant protein is expressed and farnesylated in vivo. This mutation induces extensive chemical shift perturbations in the PEX19 CTD compared to the wild type protein (Supplementary Fig. 4b), suggesting that the recognition of the farnesyl group in the

internal cavity is strongly perturbed. Enhanced hydrophobicity associated with partial exposure of the farnesyl group is indicated by hydrophobic interaction chromatography (Fig. 3a; Supplementary Fig. 4d). These data demonstrate that proper recognition of the farnesyl group by burial in the internal cavity of PEX19 plays an important role for its functional activity in vivo.

In conclusion, our study reveals an unprecedented mode of farnesyl recognition in cis by the conserved α-helical fold of the PEX19 CTD. The insight into structural and functional effects of PEX19 farnesylation represents a significant advance towards mechanistic understanding of the role of PEX19 in peroxisomal protein transport. The findings observed here for PEX19 suggest that protein farnesylation may not only affect membrane localization of proteins but may have more general and novel roles in the regulation of protein function by allosteric control of ligand binding.

## Methods

**Expression and protein purification.** Escherichia coli BL21 (DE3) (Novagen) was used for protein expression. His-tagged human PEX19 was expressed from pETM11 vector and purified as soluble protein using $Ni^{2+}$ affinity chromatography (nickel-nitrilotriacetic acid)[21]. Cell pellets were resuspended and sonicated in 30 mM Tris–HCl, pH 8.0, 300 mM NaCl, 10 mM imidazole and 1 mM β-mercaptoethanol. Lysate was centrifuged at 15,000g for 20 min, and supernatant was loaded on Ni-NTA column. Subsequently the resin was washed with 30 mM Tris–HCl, pH 8.0, 700 mM NaCl, 25 mM imidazole, 1 mM β-mercaptoethanol and protein was eluted with 30 mM Tris–HCl, pH 8.0, 150 mM NaCl, 400 mM imidazole, 1 mM β-mercaptoethanol. Hexahistidine tag was cleaved overnight by TEV protease and removed by additional $Ni^{2+}$ affinity chromatography. Flow through was loaded on size-exclusion chromatography column for final purification step. Uniformly isotope-labelled samples were obtained using M9 medium supplemented with $^{13}C$-labelled glucose and/or $^{15}NH_4Cl$, respectively. For perdeuterated proteins, M9 medium in $D_2O$ and [U- $^2H$]-D-glucose were used for protein expression. S. cerevisiae farnesyl transferase (FTase) was expressed from a bicistronic pETM11 vector (kindly provided by S. Holton, EMBL Hamburg) and purified as described for PEX19 without removing the hexahistidine tag.

**In vitro farnesylation.** In vitro farnesylation assays were carried out as described[29,30] in 50 mM Tris–HCl, pH 8.0, 20 mM KCl, 5 mM $MgCl_2$, 10 μM $ZnCl_2$ and 10 mM dithiothreitol. A reaction volume of 10 ml with 100 μM PEX19 was incubated with 130 μM farnesyl pyrophosphate (Sigma Aldrich) and 250 nM farnesyl transferase for 1 h at 37 °C. A subsequent $Ni^{2+}$ affinity chromatography (nickel-nitrilotriacetic acid) removed the FTase. The farnesylated PEX19 was further purified by size exclusion chromatography on a HiLoad 16/60 Superdex 75 (GE Healthcare) and stored in 20 mM potassium phosphate, pH 6.5 and 50 mM NaCl. The completeness of the farnesylation was analysed by SDS gel electrophoresis and $^1H$,$^{15}N$ HSQC NMR spectra.

**NMR Spectroscopy.** NMR spectra were acquired at 298 K on a Bruker Avance III 600-MHz spectrometer with a TCI cryo-probe head, an Avance III 750-MHz spectrometer with a TXI probe head, or an Avance 900 instrument with a TXI cryo-probe head. Spectra were processed using NMRPipe[31] and analysed with Sparky[32]. Backbone assignment was done semi-automatically with MARS[33].

Backbone and side chain assignments were obtained from HNCA, HNCACB, CBCA(CO)NH, H(C)CH-TOCSY and (H)CCH-TOCSY spectra[34]. [13]C- and [15]N- edited NOESY spectra with a mixing time of 70 ms were recorded to derive distance restraints. For NOEs between PEX19 and the farnesyl group, isotope-edited and filtered 3D NOESY spectra (70 and 100 ms mixing times) were acquired using specific isotope-labelled samples. [15]N $R_1$, $R_2$ relaxation rates and {[1]H}-[15]N heteronuclear NOEs were recorded at an Avance III 750 MHz spectrometer[35]. Local correlation times were derived from the [15]N $R_2/R_1$ ratio[35]. For determination of solvent paramagnetic relaxation enhancement rates, gadolinium diethylene-triamine-penta-acetic-acid bismethylamide (Gd(DTPA-BMA)) was titrated to the protein samples[36–38]. Saturation recovery [1]H,[15]N HSQC experiments with recovery times from 0.01 to 3 s were recorded at concentrations of 0, 1, 2, 3, 5, 7 and 10 mM Gd(DTPA-BMA). Peak intensities were fitted to an exponential recovery function according to equation $I(\tau) = I_0 - A\exp(-R_1\tau)$, where $\tau$ is the recovery delay, $I(\tau)$ is the peak intensity measured for the recovery delay $\tau$, $I_0$ is the maximum peak intensity, $R_1$ is the longitudinal proton relaxation rate and A is the signal amplitude. To obtain the sPRE value, the proton $R_1$ rates were collected for all measured concentrations of Gd(DTPA-BMA) $c_{para}$ and a linear regression for the equation $R_1(c_{para}) = m_{sPRE}\,c_{para} + R_1^0$ was performed, where $R_1(c_{para})$ is the proton $R_1$ measured at the concentration $c_{para}$ of the paramagnetic compound, the slope $m_{sPRE}$ corresponds to the sPRE and $R_1^0$ is the fitted proton $R_1$ in the absence of the paramagnetic compound[36]. For NMR titrations an ALDP peptide (sequence: AAKAGMNRVFLQRLL, residues 62–76 of human ALDP) was dissolved to a concentration of 5 mM in NMR buffer and added to [15]N-labelled PEX19 CTD with and without farnesylation in the same buffer up to a protein-peptide molar ratio of 1:10.

**Structure calculation.** NOE cross-peak assignments and structure calculations with torsion angle dynamics were carried out automatically with CYANA 3.0 followed by manual inspection[39]. The farnesylated cysteine was parametrized by using the Dundee PRODRG2 Server[40]. The CYANA library was prepared manually using the PDB coordinates derived from the PRODRG2 Server and all rotatable dihedral angles were defined. The dihedral angles for the isoprene double bonds were fixed to *trans* in accordance with our NMR analysis that defined the lipid as (2E,6E)-farnesol. For the last isoprenoid unit the methyls were stereospecifically assigned based on their carbon chemical shifts. Number 14 is the *trans* methyl as it has an upfield [13]C chemical shift that is in the same frequency range as found for the [13]C chemical shifts of the *trans* methyls 4 and 10 (~20 p.p.m.). Number 15 is the *cis* methyl based on its downfield [13]C chemical shift (~28 p.p.m.). For farnesyl atom numbering please see Supplementary Fig. 2a. No farnesyl-farnesyl restraints were imposed during the structure calculations. Protein-protein and protein-farnesyl NOE peak intensities were converted to upper distance limits using the internal calibration function of CYANA. Unambiguous distance restraints derived from CYANA, backbone dihedral angle restraints derived from TALOS+ (ref. 41), and residual dipolar coupling restraints were used for water refinement using CNS (ref. 42). For water refinement the parameter and topology files were taken from the PRODRG2 Server and stereochemistry was the same as in CYANA. All structures were validated with iCing (http://nmr.cmbi.ru.nl/icing/). Molecular images were generated with PyMol (Schrödinger).

**Intermolecular NOEs.** Based on initial [13]C-edited, [13]C,[15]N filtered NOESY experiments we were able to identify the methylene groups of the prolines and methyl groups in contact with the phenylalanine ring of ALDP peptide. The aromatic frequencies could be unambiguously assigned to ALDP peptide, because they were unique. Strong NOEs were also observed for ALDP methyl frequencies (Supplementary Fig. 8a) but could not be unambiguously assigned due to the large number of methyl-containing residues in the ALDP peptide. Nevertheless, these NOEs confirm that numerous contacts, involving the aromatic residues and methyl groups in the PMP ligand establish a specific complex with farnesylated PEX19. Assignments of NOEs to other ALDP residues were not possible due to overlap with residues in the farnesyl group, which was not isotope-labelled. To improve sensitivity and reduce spectral overlap PEX19 CTD was uniformly [2]H,[13]C,[15]N-labelled, except for Leu and Val methyl groups. Intermolecular NOEs were measured for complexes of this PEX19 CTD bound to the unlabelled ALDP peptide using [13]C-edited NOESY experiments[34]. From these experiments proton chemical shifts at ≈0.9 p.p.m. were unambiguously assigned to Leu182 or Leu183 (Supplementary Fig. 8a).

**Docking calculations.** On the basis of NMR titrations and NOESY experiments both ambiguous and unambiguous restraints were obtained for the complex PEX19 CTD-ALDP. The expert interface of HADDOCK docking server[25] was used to perform docking calculations. The NMR structure of farnesylated PEX19 CTD, and a helical fragment of the ALDP peptide (residues 68–76: NRVFLQRLL) were used as first and second molecule respectively. PEX19 residues with CSP > 0.05 p.p.m. (Leu172, Met175, Ser177, Asn181, Leu182, Leu183, Ser184, Lys185, Y189, L192) and the Phe71 in the ALDP peptide were used as active residues and those within 6.5 Å were treated as passive residues. In iterations 0, 1 and during water refinement 2,000, 400 and 400 structures were calculated, respectively. The minimum cluster size was set to 10. Ambiguous distance restraints were derived

from $\omega_1$-filtered/$\omega_2$-edited NOESY experiment and involve methyl protons of PEX19 Leu182 or Leu183 and methylene protons of Pro273 or Pro274 to ALDP aromatic protons of Phe71.

**Microscale thermophoresis assays.** An N-terminally fluorescein-labelled peptide LALKLRLQVLLLARV, corresponding to residues 186–200 of human PEX11B (Peptide Specialty Laboratories GmbH, Heidelberg) was dissolved to a concentration of 200 nM in 20 mM potassium phosphate, pH 6.5, 50 mM NaCl with 20% methanol and 0.1% Tween. Unlabelled PEX19 CTD with or without farnesylation was serially diluted in the same buffer to concentrations from 612 μM to 18 nM for wild type PEX19 CTD. Equal volumes of protein and peptide solutions were mixed thoroughly and centrifuged at 15,000g for 5 min. The supernatant was transferred to Microscale Thermophoresis Assays (MST) capillaries and measured with a NanoTemper Monolith NT 0.15T. Normalized fluorescence values from three separate measurements for PEX19 CTD wild type were used to determine the dissociation constants.

**Fluorescence polarization.** ALDP peptide (62-AAKAGMNRVFLQRLL-76) fused N-terminally with fluorescein isothiocyanate (FITC), was purchased from Peptide Specialty Laboratories GmbH. The lyophilized powder was dissolved in 1 ml water resulting in a 5 mM stock solution. Overnight dialysis against buffer containing 50 mM Tris pH 8.0, 50 mM NaCl, removed trifluoroacetate counterions. For the determination of each binding curve a 12 point titration was performed with constant concentration of 20 nM labelled peptide with increasing concentrations of PEX19 CTD variants. Starting concentration of PEX19 CTD was 1 mM with 12 serial dilutions reaching 0.5 μM. Reaction mixtures were then transferred into 96-well *Optiplate* and measured as triplicates in PerkinElmer EnVision plate reader.

**Hydrophobic interaction chromatography.** Purified PEX19 CTD and variants were adjusted to a buffer containing 650 mM $(NH_4)_2SO_4$ and 500 μg protein was subjected to equilibrated Butyl Sepharose FF column (20 mM potassium phosphate, pH 6.5, 50 mM NaCl and 650 mM $(NH_4)_2SO_4$). After washing, bound protein was eluted with linear decreasing $(NH_4)_2SO_4$ concentrations (650–0 mM $(NH_4)_2SO_4$, 20 ml). Eluted proteins were detected using 280 nm absorption. The binding strength is indicated by the elution volume and correlates with protein hydrophobicity.

**Peroxisomal protein import assays in fibroblasts.** A bicistronic expression plasmid coding for eGFP-PTS1 and non-tagged PEX14 (pMF1220)[43] was kindly provided by Marc Fransen (Leuven, Belgium). pIRES2-HsPEX19-eGFP-PTS1 was constructed by replacing the open reading frame of PEX14 (BglII/SalI) with a DNA-cassette encoding the full-length open reading frame of human PEX19 as derived from BamHI/SalI digestion of pAH05 (ref. 44). Single point mutations were introduced into the PEX19 sequence by Quickchange XL Site-directed Mutagenesis kit (Stratagene) using primer pairs (Supplementary Table 1). The bicistronic expression vectors coding for PEX19 variants and eGFP-PTS1 were transfected into the human cell-line ΔPEX19 T which was derived from PEX19-deficient Zellweger patient fibroblasts[21]. ΔPEX19 T cells, characterized by a defect in peroxisome biogenesis were routinely authenticated by functional complementation assays as described below and checked for mycoplasma contamination using Venor GeM Advance Mycoplasm Detection Kit (Minerva Biolabs). The efficiency of transfection according to the Amaxa Cell Line Nucleofector Kit R protocol (Lonza Cologne AG) was always around 90%. After 72 h of transfection, cells were subjected to fluorescence and immunofluorescence microscopy using polyclonal antiserum against human PEX14 (ref. 45) (1:400 diluted) or monoclonal antibodies against human PEX19 (BD Biosciences, MA5-17266 Thermo Scientific; 1:400 diluted). Statistical analysis was carried out from three independent transfections of each pIRES-PEX19-eGFP-PTS1 expression plasmid. Based on the appearance of eGFP-PTS1 fluorescence pattern, about 100 cells of each experiment were visually categorized into three classes, (i) full complementation of peroxisomal import as visualized by punctate staining pattern, (ii) partial complementation as indicated by diffuse cytosolic staining with few dots, (iii) no complementation resulting in only cytosolic staining. All micrographs were recorded on a Zeiss Axioplan 2 microscope with a Zeiss Plan-Apochromat × 63/1.4 oil objective and an Axiocam MR digital camera, and were processed with AxioVision 4.6 software (Zeiss, Jena, Germany). The steady-state level of PEX19 expression was assessed by immunoblot analysis, using monoclonal mouse antibodies against HsPEX19 (BD Biosciences, MA5-17266; 1:1,000 diluted) (Supplementary Fig. 9).

**Data availability.** UniProt accession numbers used in this study are P40855 (PEX19), P22007/P29703 (farnesyl transferase), Q92968 (PEX13), O96011 (PEX11B) and P33897 (ALDP). The atomic coordinates and restraint files for the NMR ensemble of the farnesylated PEX19 CTD are deposited in the PDB with accession code 5LNF, chemical shifts are deposited at BMRB, entry 34030. All data are available from the authors.

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

## Acknowledgements

We are grateful to A. Itzen and H. Kooshapur (TU München) for critical reading of the manuscript; S. Holton, N. Schüller, C. Williams, K. Fodor (EMBL Hamburg) for expression plasmids, material and discussions and N. Raschke (TU München), E. Becker and J. Wolf (University Bochum) for help with sample preparation and experiments. We are grateful to J. Boisbouvier, IBS Grenoble for donating the precursor for stereoselective methyl-labelling; S. Duhr and S. Blanke (NanoTemper Technologies GmbH, Munich) for support and H. Waldmann (MPI Dortmund) for spin label precursors. We thank the Bavarian NMR Centre (BNMRZ) for NMR measurement time. This work was supported by the Deutsche Forschungsgemeinschaft (SFB594, FOR1905 to M.S., ER178/4-1 to E.R.) and the EU FP7 NMI3 project (M.S.). U.S. acknowledges support from the Elitenetzwerk Bayern; K.T. from the Ministry of Education, Youth and Sports of the Czech Republic under the project CEITEC 2020 (LQ1601); T.M. from an EMBO Long Term Fellowships, a Schrödinger Fellowship by the Austrian Science Fund (FWF) and an APART-fellowship (Austrian Academy of Sciences) as well as the Bavarian Ministry of Sciences, Research and the Arts in the framework of the Bavarian Molecular Biosystems Research Network. The funders had no role in study design, data collection and analysis, decision to publish, or preparation of the manuscript.

## Author contributions

The author(s) have made the following declarations about their contributions: U.S. and L.E. performed molecular biology, protein biochemistry, NMR data acquisition and analysis and biophysical binding studies, K.T. structure calculation and NMR experiments. T.M. contributed to NMR and performed solvent PRE calculations. R.R. performed hydrophobic interaction chromatography. W.S., R.E. and J.R. designed and conducted *in vivo* analyses. M.S., W.S., R.E., U.S., K.T., M.W. and L.E. designed the project and wrote the manuscript. All authors discussed the results and commented on the manuscript.

## Additional information

**Competing financial interests:** The authors declare no competing financial interests.

**Publisher's note**: 

