## [Peer review file · Nature Communications]

Reviewer #1 (Remarks to the Author):

Nature Communications Ms. NCOMMS-16-08356-T

Title: Allosteric Modulation of Peroxisomal Membrane Protein Recognition by Farnesylation of the Peroxisomal Import Receptor PEX19

Authors: Emmanouilidis, L. et al.

Remarks:

The transport of peroxisomal membrane proteins (PMPs) requires the soluble PEX19 protein as chaperone and import receptor. Recognition of cargo PMPs by the C-terminal domain (CTD) of PEX19 is required for peroxisome biogenesis in vivo. Farnesylation at a C-terminal CAAX motif in PEX19 enhances the PMP interaction.

The authors report here that the NMR structure of the farnesylated PEX19 CTD reveals that the farnesyl moiety is buried in an internal hydrophobic cavity. This induces substantial conformational changes that allosterically reshape the PEX19 surface to form two hydrophobic pockets for the recognition of conserved aromatic/aliphatic side chains in PMP ligands. Mutations of PEX19 residues that mediate farnesyl contacts or PMP recognition abolish PMP cargo binding and cannot complement a $\Delta pex19$ phenotype in human Zellweger patient fibroblasts. The authors claim that their results demonstrate an allosteric mechanism for the regulation of protein function by farnesylation.

Comments:

Major points:

1. The resolved “solution structure” of farnesylated PEX19 CTD is the first demonstration that leads to a model on the role of farnesylation in the recognition of PMP ligands. However, two studies by the authors have already reported that farnesylation of PEX19 is required for its structural integrity, enhanced affinity to PMP ligands, and complementing activity towards *pex19* cell mutant (Rucktaschel et al., *J. Biol. Chem.*, **284**, 20885-20896 (2009), reference No. 15 in the manuscript) and crystal structure of PEX19(161-283) revealed the PMP ligand-binding domain (Schueller et al., *EMBO J.* **29**, 2491-2500 (2010), ref. 20). This study essentially demonstrated the same conclusion as in these earlier studies and just provides a reasonable explanation for the established observation of the role of PEX19 CTD in PMP recognition by structural information of farnesylated PEX19.
2. The authors claim that farnesylation of PEX19 alters the conformation of helix $\alpha 1$ at a distal position. This is based on the data showing (1) the large conformational difference between crystal structure of PEX19 CTD Δ C(161-283) and the solution structure of farnesylated PEX19 CTD(161-299) (Fig. 2, d and e) and (2) NMR analysis information of both farnesylated and non-farnesylated PEX19 CTD(161-299) in solution (Fig. 1c). The authors likely used PEX19 CTD Δ C(161-283) as a non-farnesylated form of PEX19 (page 5, lines 23, 27, 30, and 35). However, this is not adequate because PEX19 CTD Δ C(161-283) shows higher affinity to at least two PMP

ligands than non-farnesylated PEX19 CTD(161-299) (ref. 20). The affinity of PEX19 CTD Δ C(161-283) to PMP ligands is rather comparable to farnesylated PEX19 CTD(161-299) (Supplementary Fig. 5c), indicating that PEX19 CTD Δ C(161-283) is clearly different from non-farnesylated PEX19 CTD(161-299) in terms of binding to PMP ligands. Furthermore, orientation of helix α 1 in non-farnesylated PEX19 CTD Δ C(161-283) could be only the result of packing condition for crystallization. Therefore, the comparison of the helix α 1 position pointed out in Fig. 2d and 2e does not provide any logical output. On the other hand, NMR data shown in Fig. 1c indicate no substantial difference in the helix α 1 between farnesylated and non-farnesylated PEX19. More extensive data are required to evidently show the allosteric conformational change of helix α 1 induced by PEX19 farnesylation.

3. In Fig. 3, functional complementation with PEX19 was decreased to 23% by M195K mutation completely abolished by M179R and M255R mutations (Fig. 3 b and c). Why is this in vivo activity of each mutant not correlated with farnesylation-induced change in hydrophobicity (Fig. 3a) and structural information deduced by NMR analysis (Supplementary Fig. 4, a and b)? Furthermore, since there are no data regarding to how these mutations affect their binding to PMP ligands, the current data only suggest that the putative farnesyl-binding pocket is important for PEX19 function in vivo. To clarify the structure-function relationship, the authors should quantify the binding properties of these PEX19 mutants to PMP ligands.

4. On page 6, the authors claim that the farnesylation-induced increase of heteronuclear NOE values for the C-terminal residues seen for wild-type PEX19 is less pronounced in the M255R PEX19 variant (Supplementary Fig. 4e). As the authors claim, it does look less pronounced, but still looks rather significant. This indicates that significant population of the C-terminal region of the farnesylated M255R PEX19 variant adopts a conformation similar to that of farnesylated wild-type PEX19, which is more stable than the non-farnesylated form. Considering that the M255R PEX19 variant exhibits marked reduction in functional activity (Fig. 3b, c), the structure-function relationship in this regard seems still unclear. More persuasive discussion is required.

Minor points:

1. On page 5, line 22, in following five sentences, the authors use “non-farnesylated PEX19” to indicate PEX19 CTD Δ C(161-283) (page 5, lines 23, 27, 30, and 35). But, on page 5, line 37, the same word “non-farnesylated PEX19 is monomeric...” means non-farnesylated form of PEX19 CTD(161-299). As pointed out in the major point 2, properties of PEX19 CTD Δ C(161-283) and non-farnesylated PEX19 CTD(161-299) are different although both are indeed non-farnesylated form. Need to avoid misleading and to clearly distinguish between PEX19 CTD Δ C(161-283) and PEX19 CTD(161-299).

2. Figs. 3b and 3d: While the expression level of these PEX19 mutants was similar (Fig.

3d), their localization in cells may be altered due to their increasing hydrophobicity. Are a part of the PEX19 mutants detectable on peroxisomes? Intracellular localization of PEX19 mutants should be shown by immunostaining and/or subcellular fractionation.

3. On page 4, the authors claim that an increased electrophoretic mobility in SDS-PAGE (Supplementary Fig. 1b) demonstrates complete farnesylation of the PEX19 CTD. However, there is a faint band presumably corresponding to non-farnesylated PEX19 CTD in the middle lane of the SDS-PAGE gel. Thus, the word “complete” in the corresponding sentence by the authors is wrong.

4. The authors need to indicate the concentrations of PEX19 CTD and the PMP ligands used in the experiments in the legend to Supplementary Figs. 4a and 4b. As the concentrations of the samples affect binding ratio, these parameters should be clearly written.

5. On page 9, the authors describe the physiological roles of PTS1 and PEX14 to explain the concept of experiments performed in Fig. 5b. It seems better to this reviewer to mention them on page 7 where similar experiments in Fig. 3b are first mentioned.

6. On page 9, the names of several Figure panels of Fig. 5 are incorrect.

7. On one hand, on page 11, the authors discuss the role of M179 in helix $\alpha 1$ as a residue involved in direct binding to the PMP ligand. On the other hand, in the previous sections, M179 is described only as a residue constituting the farnesyl group-binding pocket. It is difficult to follow their discussion regarding this residue. Further explanation of the role of M179 in direct binding to the PMP ligand may be helpful. At least, the authors need to better describe the location of M179 relative to the PMP ligand-binding pocket, for example, in Fig. 5b.

8. The identification of a novel role of farnesylation in the affinity enhancement of a protein is interesting. However, if farnesylation is just a mean to increase the affinity of a protein with its target by about ten fold, why did the nature choose such a rather roundabout strategy for affinity enhancement? This reviewer suggests that it is helpful for the most readers if the authors could provide some persuasive hypotheses regarding this point.

Reviewers' comments:

Reviewer #2 (Remarks to the Author):

The manuscript authored by Sattler and coworkers is an interesting study aiming to gain more insight into the mechanism of how farnesylation of PEX19 may affect its function. By employing various state-of-the-art approaches (e.g., NMR spectroscopy, structure and docking calculations, microscale thermophoresis assays, mutation analysis, in cellulo complementation assays, etc.), the authors were able (i) to elucidate the solution structure of the farnesylated C-terminal peroxisomal membrane protein (PMP)-binding domain of human PEX19 (CTD), (ii) to show that CTD farnesylation induces substantial conformational changes that allosterically reshape the PEX19 surface to form two hydrophobic pockets for the recognition of conserved aromatic/aliphatic side chains in PMP ligands, and (iii) to provide evidence that mutations interfering with either farnesyl recognition or PMP interactions affect PMP binding in vitro and peroxisome biogenesis in cellulo. Based on their findings, the authors conclude that farnesylation of PEX19 regulates its function by allosteric modulation of ligand binding.

This study represents a large amount of work, the results are technically sound, and (in general) key conclusions are justified by the data. In addition, the manuscript is well-written and logically organized. However, before being published, the authors should address the following (minor) comments:

1. Page 7 (1st paragraph): "This is supported by NMR data where the most pronounced effects are observed for the M255R substitution consistent with a significant exposure of the farnesyl moiety in the M255R variant (Fig. 3a)." Fig. 3a (= Hydrophobic interaction chromatography analysis) does not show NMR data. Do the authors mean "Supplementary Fig. 4e" (= {¹H}-¹⁵N heteronuclear NOE values for PEX19 CTD M255R with and without farnesylation compared to those of the wild type protein)?
2. Page 9 (1st paragraph): "... and a diffuse background immunolabelling of the peroxisomal membrane protein PEX14" see above and correct; PEX14 (Fig. 5b, ΔPEX19)." Replace "Fig. 5b" by "Fig. 5c".
3. Page 9 (1st paragraph), Fig. 3b, and legend to Fig. 5: The authors claim that PEX14 displays a cytosolic (see 1st paragraph on page 9 and Fig. 3b) or diffuse (see legend to Fig. 5) staining pattern in PEX19-deficient cells. This information is not correct. It is well known that PEX14 is mislocalized to mitochondria in cells lacking peroxisomal remnants (and this is also what can be seen in these figures). To improve clarity and avoid confusion, the authors should correct these mistakes.
4. Based on the results shown in Fig. 3c, the authors claim that peroxisomal protein import is partially impaired in the absence of farnesylation (compare the PEX19 and C296S conditions). However, the observed differences are minor, and it is not clear whether or not these differences are statistically significant (despite the fact that the authors state on page 15 that statistical analysis was carried out). In case there is indeed a significant difference, another problem arises. Indeed, the values shown for PEX19 in Fig. 5d (according to text on page 9, 1st paragraph, line 12: 50%) are

comparable to those shown for C296S in Fig. 3c (according to text on page 7, paragraph 2, line 5: 46%). How do the authors explain this internal inconsistency? In this context, it is important for the authors to realize that it has been reported in literature that overexpression of PEX19 retains PMPs in the cytosol (and this may in turn interfere with PTS1 protein import). As such, it is not only important to compare the expression levels of the overexpressed recombinant proteins (see Figs. 3d and 5e), but also to determine (and list) the transfection efficiencies (both factors are key determinants to obtain an idea about the amount of PEX19 per cell). Therefore, the authors should provide this information for all conditions shown in Figs. 3c and 5d. In addition, they should mention in the legend whether the vertical bars represent standard deviations or standard errors.

5. Page 10 (second last sentence before the discussion): "Cells, which express the PEX19 P273F/P274F mutant, displayed a significantly lower complementation rate of only 20 % (Fig. 5c), independent of the PEX19 protein level (Fig. 5d)." Replace "Fig. 5c" and "Fig. 5d" by "Fig. 5d" and "Fig. 5e", respectively.

6. Page 10 (last sentence before the discussion): "The impaired function of the PEX19 mutant demonstrates that the lid region is not only critical for binding of the half-transporter ALDP but also for other PMPs, in particular for those which are essential for biogenesis of peroxisomes." As peroxisome biogenesis is only reduced from 50% (upon complementation with PEX19) to 20% (upon complementation with PEX19 P273F/P274F), the term "critical" may be a little bit overstated.

8. The authors present in Fig. 6 a schematic model for the allosteric regulation of PMP recognition by farnesylation of PEX19. However, as (i) farnesylation is an irreversible process, and (ii) virtually all PEX19 is farnesylated in cellulo (see Fig. 3d), this would imply that virtually all PEX19 in a cell would be present in the closed conformation. As such, it can be questioned how farnesylation may allosterically "regulate" or "modulate" PEX19 function. Do the authors have any idea whether or not farnesylation of PEX19 influences the on- and/or off-rates of PMP binding? Also, given that (i) in cellulo, the CAAX box does not only direct the covalent attachment of the farnesyl group to the cysteine in the CAAX motif, but also the proteolysis of the three C-terminal (AAX) amino acids and the methylation of the newly exposed farnesylcysteine residue, and (ii) the latter process is reversible, it would be nice if the authors could provide a calculated guess (e.g., in the discussion) about how methylation and demethylation of the farnesylcysteine residue may affect PMP recognition.

9. Legend to supplementary Fig. 2: "To resolve ambiguities for key residues that are in contact with the farnesyl group an optimized isotope labeling strategy was used combining amino acid-selective labeling, specific methyl labeling [44], and reverse labeling of phenylalanines [45]." The reference numbers should be [45] and [46].

Reviewer #3 (Remarks to the Author):

* Overall comments

In this manuscript, the authors perform solution NMR spectroscopy studies complemented with mutagenesis and cellular functional assays to determine the mechanism by which farnesylation of PEX19 affects the binding of peptides derived from peroxisomal membrane proteins (PMPs) and cellular function (import of a peroxisomal marker protein into peroxisomes).

A crystal structure of a truncated PEP19 C-terminal domain (CTD) lacking the farnesylation site located in the extreme C-terminus was previously reported. Here, the authors show that while the extreme C-terminus of PEP19 is conformationally flexible, farnesylation induces a significant stabilization of this region. Using NMR methods, the authors determined the solution structure of the farnesylated PEP19 CTD. When compared to the truncated CTD crystal structure, the NMR solution structure showed a significant conformational rearrangement of a particular helix (helix a1) and lid region that pack against the farnesyl group, which is interestingly buried inside the structure. Using a mutagenesis and a cellular complementation assay, the authors show that residues contacting the farnesyl group are important for PEP19 function.

Next, the authors performed NMR analysis to determine where peptides derived from PMPs interact with PEX19. These studies revealed the site of interaction to involve helix a1. Guided by additional NMR data, the program HADDOCK was used to obtain a docked complex of PEX19 CTD with a PMP peptide, which revealed interactions with helix a1 and the lid region. Using mutagenesis and a cellular complementation assay, the authors demonstrated the functional importance of this PMP peptide interaction surface in PEX19-mediated import of PMPs into peroxisomes.

Overall, this is a well written manuscript that describes complementary structure-function studies to provide important new insight into the role of farnesylation in the function of PEX19. The quality of the NMR studies in particular is strong. As discussed below in my specific comments, including some additional detail would strengthen various parts of the manuscript.

* Specific comments

It would be helpful if PEX19 was defined as "Peroxisomal biogenesis factor 19" somewhere early in the manuscript.

Abstract: perhaps the authors could add a transition sentence or statement before "The NMR structure of ..." about what is unknown and how the current study fills this gap in knowledge.

Introduction (bottom of page 2): the authors allude to the fact that the "molecular mechanisms of matrix protein recognition for transport into peroxisomes are well characterized (ref 3)". Perhaps a brief, one sentence summary could be added to inform the readers?

Introduction (bottom of page 3): PEX19 is stated to bind PMP ligands. A "ligand" can mean different

things to different people. Is it a peptide, protein domain, or small molecule ligand?

Supplementary Figure 1c/d: if TALOS was used to determine secondary structure plots above the d(Ca-Cb) plot, or if this represents the crystal structure secondary structure, etc. -please add this detail to the legend.

Page 4: farnesylation of PEX19 is said to not only induce a stabilizing conformational change near the site of farnesylation, but also the N-terminal region of helix a1. However, there seems to be barely any detectable increase in hetNOE and decrease in sPRE for the N-terminal region of helix a1 in Figure 1c.

Page 5, re: second sentence: it is not clear how Supplementary Fig. 2a supports the statement that isotope-filtered NOESY experiments allowed unambiguous assignment of chemical shifts and NOEs. The supplemental legend indicates NOESY spectra would be shown, but instead only a plot of the # of NOE contacts to specific farnesyl atoms is shown. Also, it would be of general interest to the readers to understand how difficult or easy it was to assign farnesyl proton chemical shifts, how this was performed, the degree of chemical shift overlap between farnesyl resonances, and example NOESY spectra for related atoms (e.g. methyls at positions 4, 10, and 14; other aliphatic protons at the other positions).

Page 5: are protein-farnesyl NOEs observed for all specific interactions shown in Figure 2c? If yes, perhaps this should be stated (and if possible NOE connections plotted onto a structure for comparison, perhaps as a supplemental figure panel).

Page 5: it is stated the NMR data show a distinct bent conformation of the isoprenoid. In the NMR structure calculations, were any specific restraints placed on the farnesyl group to ensure a specific conformation, for example? No detail is included in the CYANA structure calculations for how a farnesyl group was incorporated into the structure calculations.

Supplementary Figure 3: PDB codes should be listed for the structures used to compare to the farnesyl conformations. In particular, were any of these NMR-derived structures? If not, and if the compared structures were all derived from x-ray crystallography, could it be that the bent conformation could be an artifact of the NMR structure calculation? Are the farnesyl groups in the structures shown in this supplementary figure buried or solvent exposed (it's difficult to tell from the figure)?

Page 5 and Figure 2e: despite the conformational rearrangement of the "lid" region, there is very little change in hetNOE and sPRE (Figure 1c). There could be some change in R1 or R2, but it is difficult to compare the relaxation parameters in Supplemental Figure 1c,d. Could a different plot be added to illustrate any dynamical differences in the lid region, or other regions in general, that occurs upon farnesylation?

Page 6: the text describing residues mutated to impair interactions the farnesyl moiety refer to Figure 2b, but I think it would be more appropriate to refer to Figure 2c.

Page 6: how was it confirmed the PEX19 mutants were completely farnesylated in vitro?

Supplementary Figure 4a/b: it would be helpful if NMR spectral overlays were shown for each mutant with or without farnesylation.

Supplementary Figure 4c/d: it would be helpful if wild-type PEX19 was shown for comparison. Also, were these plots generated for wild-type vs. mutant? This is not specified in the legend.

Supplementary Figure 4f: it would be helpful if the difference volume listed under the chromatograms were labeled as such in the actual figure (or using a greek delta symbol). Otherwise this could be confused by the reader as an x-axis volume tick mark.

Figure 3a: how were the relative differences in hydrophobicity calculated? Is the unit of the y-axis a volume (mL)?

Page 7: before discussing the functional complementation assay data (Figure 3b/c), it would be useful for the readers if a brief description is added after the second sentence about what this assay measures/detects and how this is related to PEX19 known function(s). Also, in Figure 3b/c the effect of wild-type vs. mutants in rescuing the (Δ)pex19 phenotype is shown, but I would think a control (empty plasmid/vector) should also be shown to set the baseline that these proteins rescue the phenotype, similar to what is shown in Figure 5c.

Figure 4 and Supplementary Figure 5: while in general the data show that three different PMP peptides can bind the PEX19 CTD via NMR and microscale thermophoresis (MST), there are some inconsistencies between the experiments and addressing these would strengthen this section. For example, in Figure 4 the binding of a PMP peptide from ALDP is shown by NMR at a 5x and 10x molar equivalent added, but no MST data or analysis of NMR chemical shift perturbations is shown to confirm the binding affinity. In Supplementary Figure 5a, NMR data is shown for a PMP peptide from PEX13 at a 1.5x molar equivalent added, but in 5b the analysis is performed with a 0.5x molar equivalent added and again there is no analysis of binding affinity. Here, knowing the affinity would provide insight into the apparent intermediate exchange binding profile in 5a (right panel). In contrast to the previous two PMP peptides, Supplementary Figure 5c shows MST binding affinity data a PMP peptide derived from PEX11b, but no NMR data are shown to map the interaction surface on the PEX19 CTD.

It is stated in the abstract (and the discussion) that farnesylation orchestrates the PMP binding region of PEX19, which includes helix α 1, by an allosteric mechanism. However, while the farnesylation site is far removed from the PMP binding region in sequence space, it is nearby in 3D/structural space. Is this really a (long-range) allosteric mechanism, or is it merely that farnesylation directly stabilizes this region because of the nearby proximity to helix α 1-and therefore facilitates PMP peptide binding to the same surface? What does allosteric mean here?

Farnesylation is known (or at least suggested) to play a role in trafficking proteins to membrane surfaces, and PEX19's ability to bind peroxisomal membrane proteins (PMPs) may suggest it could tether to a membrane. Here, farnesylation is shown to affect PMP interaction, but nothing is

discussed about whether PEX19 can interact with membranes, and if it can, the possible role of the (buried) farnesyl group and the mechanism by which this it may become solvent accessible if that is required for membrane interaction?

For the HADDOCK PEX19/ALDP PMP complex, it would be useful if the "HADDOCKSCORE vs. i-RMSD" figure plot is shown to complement the data provided in Supplemental Table S2.

Just a reminder that PDB and BMRB accession codes should be added prior to publication when obtained.

Point-by-point response

We thank all reviewers for the constructive and helpful comments. We have performed a number of additional experiments and provide new data in the revised manuscript, which, together with revisions in the text, address the points raised.

Please note that supplementary figure numbers have changed compared to the original submitted version, below we refer to the figure numbers in the revised manuscript.

Reviewer #1

Major points:

1. The resolved “solution structure” of farnesylated PEX19 CTD is the first demonstration that leads to a model on the role of farnesylation in the recognition of PMP ligands. However, two studies by the authors have already reported that farnesylation of PEX19 is required for its structural integrity, enhanced affinity to PMP ligands, and complementing activity towards pex19 cell mutant (Rucktaschel et al., J. Biol. Chem., 284, 20885-20896 (2009), reference No. 15 in the manuscript) and crystal structure of PEX19(161-283) revealed the PMP ligand-binding domain (Schueller et al., EMBO J. 29, 2491-2500 (2010), ref. 20). This study essentially demonstrated the same conclusion as in these earlier studies and just provides a reasonable explanation for the established observation of the role of PEX19 CTD in PMP recognition by structural information of farnesylated PEX19.

Thank you for the appreciation of our work. We would like to stress that none of the two earlier papers mentioned, provided any molecular or structural data to rationalize the role of farnesylation for PMP ligand binding. We would argue that our data go far beyond published data, and rather show a completely unexpected mode of farnesyl burial and allosteric molecular mechanism for the enhanced PMP binding.

We also like to clarify that Schueller et al have not established molecular mechanisms of PMP binding. In this study hydrophobic residues in helix $\alpha 1$ were introduced to disrupt the helical conformation, but not identifying residues critical for PMP binding. Moreover, the orientation of helix $\alpha 1$ in the crystal structure is induced by oligomerization in the crystal and does not reflect the solution conformation (see below). We are therefore convinced that our data significantly extend the previous work, which did not reveal any specific structural details regarding PMP recognition.

2. The authors claim that farnesylation of PEX19 alters the conformation of helix $\alpha 1$ at a distal position. This is based on the data showing (1) the large conformational difference between crystal structure of PEX19 CTD Δ C(161-283) and the solution structure of farnesylated PEX19 CTD(161-299) (Fig. 2, d and e) and (2) NMR analysis information of both farnesylated and non-farnesylated PEX19 CTD(161-299) in solution (Fig. 1c). The authors likely used PEX19 CTD Δ C(161-283) as a non-farnesylated form of PEX19 (page 5, lines 23, 27, 30, and 35).

The assumption is not correct. Our NMR data compare the effect of farnesylation and flexibility of the CaaX box residues for the construct 161-299 with and w/o farnesylation. For the structural comparison we indeed use the solution structure of farnesylated PEX19 CTD(161-299) and the crystal structure of the truncated non-farnesylated PEX19 CTD Δ C(161-283). We went through the manuscript to clearly state in the relevant parts of the manuscript which proteins were used to avoid any misunderstanding.

However, this is not adequate because PEX19 CTD Δ C(161-283) shows higher affinity to at least two PMP ligands than non-farnesylated PEX19 CTD(161-299) (ref. 20). The affinity of PEX19 CTD Δ C(161-283) to PMP ligands is rather comparable to farnesylated PEX19 CTD(161-299) (Supplementary Fig. 5c), indicating that PEX19 CTD Δ C(161-283) is clearly different from non-farnesylated PEX19 CTD(161-299) in terms of binding to PMP ligands.

We would like to point out that all experiments presented in our manuscript have been performed with the construct 161-299 with and w/o farnesylation. We agree that non-farnesylated CTD Δ C(161-283) may be different in PMP binding from non-farnesylated CTD (161-299), but we would consider that this is less relevant as CTD Δ C is a truncated form that lacks the CAAX box and is not present in cells. However, we do note that our NMR analysis of non-farnesylated CTD (161-299) is not compatible with the orientation of helix $\alpha 1$ in the crystal structure of non-farnesylated CTD Δ C(161-283) which is attributed to crystal packing induced by the tetramerization of non-farnesylated CTD Δ C(161-283). We have not performed structural or functional analysis of non-farnesylated CTD Δ C(161-283) in solution because we consider

that the biological functional form of the protein is the one that harbors the CAAX motif, i.e. non-farnesylated CTD (161-299).

In this respect, we have compared the affinities of PEX19 161-299, i.e. the same protein construct using the same assay with and w/o farnesylation and found that farnesylation increases the affinity to different PMP ligands several folds. Our structural analysis demonstrates that the residues that mediate PMP contacts are present in the shorter CTD (161-283). However, a key finding from our NMR and structural data is that farnesylation allosterically reshapes the PMP binding surface and thereby explains the enhanced affinity.

As we stated (and noted by the reviewer) it has been previously reported (Rucktaschel et al., J. Biol. Chem, 2009) that farnesylation enhances the PMP interaction. We have tested three different peptides using NMR titration, MST and FP experiments, and found that these peptides exhibit different affinities depending on the specific PMP and peptide region studied (Supplementary Fig. 6, Supplementary Fig. 7). The different affinities are not surprising as they refer to different PMPs and regions studied but in all cases a notably improved affinity is observed upon farnesylation.

Furthermore, orientation of helix $\alpha 1$ in non-farnesylated PEX19 CTD Δ C(161-283) could be only the result of packing condition for crystallization. Therefore, the comparison of the helix $\alpha 1$ position pointed out in Fig. 2d and 2e does not provide any logical output. On the other hand, NMR data shown in Fig. 1c indicate no substantial difference in the helix $\alpha 1$ between farnesylated and non-farnesylated PEX19. More extensive data are required to evidently show the allosteric conformational change of helix $\alpha 1$ induced by PEX19 farnesylation.

We fully agree and, in fact, have stated in the text that the orientation of helix $\alpha 1$ reported by Schueller et al is artificial and does not reflect the solution conformation. The crystal structure of CTD Δ C(161-283) is the only structure available to compare with, so we used this as a reference nevertheless. Importantly, we describe and show based on our NMR data (Figure 1c and Supplementary Fig. 1c,d) that in the non-farnesylated CTD helix $\alpha 1$ is pre-formed, but flexible in solution and not rigidly packed against the globular fold of the CTD. Unfortunately, due to signal overlap and line-broadening we could not completely assign and analyze all NMR signals in helix $\alpha 1$. Therefore, a more high-resolution description is not feasible. The flexibility and line-broadening observed in the NMR spectra suggests that helix $\alpha 1$ is transiently interacting in a dynamic way with the globular CTD fold. This flexibility may also rationalize why crystallization induces an artificial packing of this helix, as the flexibility otherwise would interfere with crystal packing.

3. In Fig. 3, functional complementation with PEX19 was decreased to 23% by I195K mutation completely abolished by M179R and M255R mutations (Fig. 3 b and c). Why is this *in vivo* activity of each mutant not correlated with farnesylation-induced change in hydrophobicity (Fig. 3a) and structural information deduced by NMR analysis (Supplementary Fig. 4, a and b)? Furthermore, since there are no data regarding to how these mutations affect their binding to PMP ligands, the current data only suggest that the putative farnesyl-binding pocket is important for PEX19 function *in vivo*. To clarify the structure-function relationship, the authors should quantify the binding properties of these PEX19 mutants to PMP ligands.

Thank you for this valuable suggestion. As requested we have measured binding affinities for the WT and mutant PEX19 CTD using a fluorescence polarization assay. All mutants show impaired binding to ALDP peptide *in vitro* (Supplementary Fig. 6), supporting further the functional deficiency *in vivo*.

We believe that our *in vitro* and *in vivo* data are indeed consistent with the change in hydrophobicity. All mutants show increased hydrophobicity indicating that the farnesyl is not properly buried. In support our NMR analysis shows that the mutations induce chemical shift perturbations across the PEX19 CTD suggesting that the protein conformation is somewhat altered (Supplementary Fig. 4a and b). We also note that M179 and I195 not only contribute to farnesyl binding but mediate direct contacts to the PMP ligand, whereas M255R only affects the recognition of the farnesyl group and therefore has an indirect effect on the PMP interaction. This mutant thus indicates that the proper recognition of the farnesyl group in the cavity of the PEX19 CTD is required to shape the binding surface for the PMP ligands.

Please note that our farnesyl binding pocket is not “putative”, as we report a high-resolution and well-defined NMR structure that unambiguously shows recognition of the farnesyl group in the internal cavity of the Pex19 CTD.

4. On page 6, the authors claim that the farnesylation-induced increase of heteronuclear NOE values for the C-terminal residues seen for wild-type PEX19 is less pronounced in the M255R PEX19 variant (Supplementary Fig. 4e). As the authors claim, it does look less pronounced, but still looks rather significant. This indicates that significant population of the C-terminal region of the farnesylated M255R PEX19 variant adopts a conformation similar to that of farnesylated wild-type PEX19, which is more stable than the non-farnesylated form. Considering that the M255R PEX19 variant exhibits marked reduction in functional activity (Fig. 3b, c), the structure-function relationship in this regard seems still unclear. More persuasive discussion is required.

We have extended the discussion of these data on p.6 in the revised manuscript.

Note, that the heteronuclear NOE data report on internal motion at sub-nanosecond timescales and not directly on changes in conformation. The reduced heteronuclear NOE value of the farnesylated M255R CTD is significant as a value below 0.6 is typical for flexible N- or C-terminal regions in proteins.

The fact that the tail does not become completely flexible may reflect 1) that a minor population of molecules exists where the farnesyl group is not completely excluded from the cavity and/or 2) that the farnesyl group may weakly bind to the surface of the Pex19 CTD (even when it is excluded), thus explaining the reduction in mobility: The farnesyl group is water insoluble. Once it is transferred to PEX19 it is immediately shielded from the solvent in the protein cavity. As a consequence the C-terminus of the protein becomes rigid, as documented by our heteronuclear NOE data. M255 lies at the heart of the farnesyl cavity and was replaced by arginine, with the aim to impair farnesyl burial and the function of PEX19 *in vivo*. Our analysis on the hydrophobicity of this mutant shows that the farnesyl is not properly buried and partly exposed. It appears that the mutation results in distinct hydrophobic interactions with the farnesyl that significantly affect the protein conformation and its function. The hydrophobic network of interactions between farnesyl and protein is not optimal in the M255R mutant, thus explaining the reduced mobility of the C-terminal region. A flexible C-terminal region for the farnesylated M255R, as seen for the non-farnesylated proteins, would imply that the farnesyl does not interact with the protein but in this case we would not be able to obtain soluble protein for our analysis (NMR, binding assays, hydrophobicity analysis). We also note that the M255R mutant was not stable for long time in solution after purification.

Minor points:

1. On page 5, line 22, in following five sentences, the authors use “non-farnesylated PEX19” to indicate PEX19 CTDAC(161-283) (page 5, lines 23, 27, 30, and 35). But, on page 5, line 37, the same word “non-farnesylated PEX19 is monomeric...” means non-farnesylated form of PEX19 CTD(161-299). As pointed out in the major point 2, properties of PEX19 CTDAC(161-283) and non-farnesylated PEX19 CTD(161-299) are different although both are indeed non-farnesylated form. Need to avoid misleading and to clearly distinguish between PEX19 CTDAC(161-283) and PEX19 CTD(161-299).

We apologize for these inconsistencies and have corrected this in the revised text.

2. Figs. 3b and 3d: While the expression level of these PEX19 mutants was similar (Fig.3d), their localization in cells may be altered due to their increasing hydrophobicity. Are a part of the PEX19 mutants detectable on peroxisomes? Intracellular localization of PEX19 mutants should be shown by immunostaining and/or subcellular fractionation.

As requested by the reviewer, we tested the intracellular localization of Pex19 wild-type as well as all single and double mutants used in this study by immunofluorescence microscopy. The mutations which increase the hydrophobicity of PEX19 did not result in stronger association with peroxisomes or significant mistargeting to other subcellular compartments. The results are shown in the new Supplementary Fig. 5 and described in the main text on page 7.

3. On page 4, the authors claim that an increased electrophoretic mobility in SDS-PAGE (Supplementary Fig. 1b) demonstrates complete farnesylation of the PEX19 CTD. However, there is a faint band presumably corresponding to non-farnesylated PEX19 CTD in the middle lane of the SDS-PAGE gel. Thus, the word “complete” in the corresponding sentence by the authors is wrong.

We have confirmed complete farnesylation by mass spectrometry when we implemented the farnesylation protocol. In addition, our NMR spectra demonstrate that farnesylation is well above 95% as no NMR signals are observed that would correspond to the non-farnesylated form (Fig. 1b). We have changed the statement to “virtually complete”, but in any case believe that this level of farnesylation is sufficient.

4. The authors need to indicate the concentrations of PEX19 CTD and the PMP ligands used in the experiments in the legend to Supplementary Figs. 4a and 4b. As the concentrations of the samples affect binding ratio, these parameters should be clearly written.

We have added the protein and ligand concentrations/ratios to the legend in the revised Suppl. Fig. 4 and Suppl Fig 7.

5. On page 9, the authors describe the physiological roles of PTS1 and PEX14 to explain the concept of experiments performed in Fig. 5b. It seems better to this reviewer to mention them on page 7 where similar experiments in Fig. 3b are first mentioned.

Thank you for this suggestion, we have moved the explanation as suggested.

6. On page 9, the names of several Figure panels of Fig. 5 are incorrect.

We apologize for this, the names have been corrected.

7. On one hand, on page 11, the authors discuss the role of M179 in helix D1 as a residue involved in direct binding to the PMP ligand. On the other hand, in the previous sections, M179 is described only as a residue constituting the farnesyl group-binding pocket. It is difficult to follow their discussion regarding this residue. Further explanation of the role of M179 in direct binding to the PMP ligand may be helpful. At least, the authors need to better describe the location of M179 relative to the PMP ligand-binding pocket, for example, in Fig. 5b.

We have clarified the discussion of the role of M179 and indicate the location of this residue in Fig. 5b as suggested.

8. The identification of a novel role of farnesylation in the affinity enhancement of a protein is interesting. However, if farnesylation is just a mean to increase the affinity of a protein with its target by about ten fold, why did the nature choose such a rather roundabout strategy for affinity enhancement? This reviewer suggests that it is helpful for the most readers if the authors could provide some persuasive hypotheses regarding this point.

We agree that farnesylation likely plays additional roles for the function of Pex19, but have restrained from an extensive discussion of such roles. We have now added a few sentences discussing the possible role of the farnesyl group in regulating interactions of Pex19 at the membrane (p.11). For example, it is conceivable that release of the PMP ligand at the membrane upon contact with its docking protein Pex3 could allow the farnesyl group to be extruded from the cavity and provide additional interactions with the membrane. Additional experiments in future studies will be needed to test this hypothesis.

We also have added a short discussion (p. 11) on potential carboxymethylation as further processing of the farnesylated protein (see response to reviewer 2).

Reviewer #2

1. Page 7 (1st paragraph): "This is supported by NMR data where the most pronounced effects are observed for the M255R substitution consistent with a significant exposure of the farnesyl moiety in the M255R variant (Fig. 3a)." Fig. 3a (= Hydrophobic interaction chromatography analysis) does not show NMR data. Do the authors mean "Supplementary Fig. 4e" (= {1H}-15N heteronuclear NOE values for PEX19 CTD M255R with and without farnesylation compared to those of the wild type protein)?

The inconsistency was corrected. "NMR data" was replaced by "hydrophobic interaction chromatography analysis"

2. Page 9 (1st paragraph): "... and a diffuse background immunolabelling of the peroxisomal membrane protein PEX14" see above and correct; PEX14 (Fig. 5b, ΔPEX19)." Replace "Fig. 5b" by "Fig. 5C".

The inconsistency was corrected.

3. Page 9 (1st paragraph), Fig. 3b, and legend to Fig. 5: The authors claim that PEX14 displays a cytosolic (see 1st paragraph on page 9 and Fig. 3b) or diffuse (see legend to Fig. 5) staining pattern in PEX19-deficient cells. This information is not correct. It is well known that PEX14 is mislocalized to mitochondria in cells lacking peroxisomal

remnants (and this is also what can be seen in these figures). To improve clarity and avoid confusion, the authors should correct these mistakes.

We have made the corrections as suggested. The sentence on page 9 now reads “Cells that are not complemented by functional PEX19 show a cytosolic mislocalization of the peroxisomal marker protein GFP-PTS1 and a diffuse background immunolabelling of the peroxisomal membrane protein PEX14 that is different from the peroxisomal pattern and known to be due to mislocalization to mitochondria (Fig. 5c, ΔPEX19).” Accordingly, the legend of Figure 5 reads “The same plasmid lacking PEX19 was used as a negative control (ΔPEX19) showing diffuse staining due to mitochondria mislocalization for both peroxisomal marker proteins, eGFP-PTS1 (matrix protein) and PEX14 (PMP).”

The mitochondrial mislocalization of Pex14 in PEX19-deficient and non-complemented cells, which is also visible in Fig. 3b, has been described on page 7 with the following sentence. “The same plasmid lacking PEX19 was used as a negative control (ΔPEX19) showing cytosolic and mitochondrial mislocalization for the peroxisomal marker proteins eGFP-PTS1 (matrix protein) and PEX14 (PMP), respectively.”

4. Based on the results shown in Fig. 3c, the authors claim that peroxisomal protein import is partially impaired in the absence of farnesylation (compare the PEX19 and C296S conditions). However, the observed differences are minor, and it is not clear whether or not these differences are statistically significant (despite the fact that the authors state on page 15 that statistical analysis was carried out). In case there is indeed a significant difference, another problem arises. Indeed, the values shown for PEX19 in Fig. 5d (according to text on page 9, 1st paragraph, line 12: 50%) are comparable to those shown for C296S in Fig. 3c (according to text on page 7, paragraph 2, line 5: 46%). How do the authors explain this internal inconsistency? In this context, it is important for the authors to realize that it has been reported in literature that overexpression of PEX19 retains PMPs in the cytosol (and this may in turn interfere with PTS1 protein import). As such, it is not only important to compare the expression levels of the overexpressed recombinant proteins (see Figs. 3d and 5e), but also to determine (and list) the transfection efficiencies (both factors are key determinants to obtain an idea about the amount of PEX19 per cell). Therefore, the authors should provide this information for all conditions shown in Figs. 3c and 5d. In addition, they should mention in the legend whether the vertical bars represent standard deviations or standard errors.

We agree with the reviewer that the effect of the lack of the farnesyl-moiety is not very drastic, but consider this still significant (in agreement with published data (Sacksteder et al, Gould J Cell Biol 2000)). In this first study about human PEX19 with a mutation of the farnesylation site, a reduction of PEX19 activity in a comparable range was observed. Whereas expression of WT PEX19 led to rescue of peroxisome biogenesis in ~50% of the cells, expression of PEX19/C296A reduced functional complementation to ~80% of the activity of wild-type PEX19. We compared cell phenotypes within the same experiment series. Thus, for the series shown in Fig. 3C, 60% of the cells which received wild-type PEX19 showed reconstitution of normal peroxisomal import. Thus, the 50% import seen upon transfection of the C296S mutant is significant. A cross-comparison of the results shown in Fig. 3c and 5d might be misleading as different passages of cell lines have been used.

We used the Amaxa Cell Line Nucleofactor Kit R with transfection efficiency of about 90%. Accordingly, the immunoblot can serve as a measure for a comparison of the expression level. We added the missing information to the Methods section as requested. Data shown represent mean ± standard deviation (SD) from three independent experiments.

5. Page 10 (second last sentence before the discussion): "Cells, which express the PEX19 P273F/P274F mutant, displayed a significantly lower complementation rate of only 20 % (Fig. 5c), independent of the PEX19 protein level (Fig. 5d)." Replace "Fig. 5c" and "Fig. 5d" by "Fig. 5d" and "Fig. 5e", respectively.

The inconsistency was corrected.

6. Page 10 (last sentence before the discussion): "The impaired function of the PEX19 mutant demonstrates that the lid region is not only critical for binding of the half-transporter ALDP but also for other PMPs, in particular for those which are essential for biogenesis of peroxisomes." As peroxisome biogenesis is only reduced from 50% (upon complementation with PEX19) to 20% (upon complementation with PEX19 P273F/P274F), the term "critical" may be a little bit overstated.

“Critical” was replaced by “important”.

8. The authors present in Fig. 6 a schematic model for the allosteric regulation of PMP recognition by farnesylation

of PEX19. However, as (i) farnesylation is an irreversible process, and (ii) virtually all PEX19 is farnesylated in cellulo (see Fig. 3d), this would imply that virtually all PEX19 in a cell would be present in the closed conformation. As such, it can be questioned how farnesylation may allosterically "regulate" or "modulate" PEX19 function. Do the authors have any idea whether or not farnesylation of PEX19 influences the on- and/or off-rates of PMP binding? Also, given that (i) in cellulo, the CAAX box does not only direct the covalent attachment of the farnesyl group to the cysteine in the CAAX motif, but also the proteolysis of the three C-terminal (AAX) amino acids and the methylation of the newly exposed farnesylcysteine residue, and (ii) the latter process is reversible, it would be nice if the authors could provide a calculated guess (e.g., in the discussion) about how methylation and demethylation of the farnesylcysteine residue may affect PMP recognition.

Thank you for this excellent comment. We agree that the farnesylation is not regulated and rather that the proteolysis and methylation of the CAAX motif maybe be a mechanism to regulate, as suggested by the reviewer. Studying this mechanisms will be done in future work and is beyond the scope of the current manuscript. However, we have added text discussing this possibility and its implications on p.11 in the revised manuscript.

9. Legend to supplementary Fig. 2: "To resolve ambiguities for key residues that are in contact with the farnesyl group an optimized isotope labeling strategy was used combining amino acid-selective labeling, specific methyl labeling [44], and reverse labeling of phenylalanines [45]." The reference numbers should be [45] and [46].

Thank you for this comment, we have corrected this according to the guidelines of the journal and listed in the supplement as Supplementary References numbered 1 and 2

Reviewer #3

Thank you for the overall appreciation of our work and the constructive and helpful suggestions, which we have addressed as indicated below.

It would be helpful if PEX19 was defined as "Peroxisomal biogenesis factor 19" somewhere early in the manuscript.

PEX19 now introduced as "Peroxisomal biogenesis factor 19" on page 3, as suggested.

Abstract: perhaps the authors could to add a transition sentence or statement before "The NMR structure of ..." about what is unknown and how the current study fills this gap in knowledge.

The abstract has been edited as suggested.

Introduction (bottom of page 2): the authors allude to the fact that the "molecular mechanisms of matrix protein recognition for transport into peroxisomes are well characterized (ref 3)". Perhaps a brief, one sentence summary could be added to inform the readers?

More information regarding matrix protein recognition was added.

Introduction (bottom of page 3): PEX19 is stated to bind PMP ligands. A "ligand" can mean different things to different people. Is it a peptide, protein domain, or small molecule ligand?

"PMP ligands" has been changed to "PMPs".

Supplementary Figure 1c/d: if TALOS was used to determine secondary structure plots above the d(Ca-Cb) plot, or if this represents the crystal structure secondary structure, etc.-please add this detail to the legend.

We have clarified this in the legend of Suppl. Fig. 1. Both NMR-derived secondary chemical shifts and the secondary structure in the previous crystal structure agree with each other. The last sentence of the figure legend now reads "Secondary structure elements, derived from the NMR structure for the farnesylated PEX19 and from the crystal structure for the non-farnesylated PEX19, are indicated on top together with the farnesylation site. Structures and C^α/C^β chemical shifts for both forms of PEX19 agree that the helical content is not altered by farnesylation."

Page 4: farnesylation of PEX19 is said to not only induce a stabilizing conformational change near the site of farnesylation, but also the N-terminal region of helix a1. However, there seems to be barely any detectable increase in hetNOE and decrease in sPRE for the N-terminal region of helix a1 in Figure 1c.

We would like to thank the reviewer for noticing these significant details. Our secondary chemical shift analysis shows that helix $\alpha 1$ is present irrespective of whether PEX19 is farnesylated or not and in agreement with the NMR and crystal structures (Supplementary Fig. 1). Since the backbone $\{^1\text{H}\}$ - ^{15}N heteronuclear NOE provides information about the motion of individual N-H bond vectors these values do not differ for helix $\alpha 1$. However, in the sPRE data – at least for the amides that we could analyze – there is solvent protection observed in the farnesylated PEX19 consistent with the structure where helix $\alpha 1$ adopts a defined orientation and forms part of the farnesyl cavity. We have corrected the following sentences in page 4 to state this more clearly: “*Similar sPRE variations, although to a lesser extent, are also observed for residues in the N-terminal helix $\alpha 1$ (Fig. 1c). These data demonstrate that farnesylation induces substantial conformational changes in the PEX19 CTD that primarily stabilizes the C-terminal region but also determines the relative orientation of helix $\alpha 1$ (see below).*”

Page 5, re: second sentence: it is not clear how Supplementary Fig. 2a supports the statement that isotope-filtered NOESY experiments allowed unambiguous assignment of chemical shifts and NOEs. The supplemental legend indicates NOESY spectra would be shown, but instead only a plot of the # of NOE contacts to specific farnesyl atoms is shown. Also, it would be of general interest to the readers to understand how difficult or easy it was to assign farnesyl proton chemical shifts, how this was performed, the degree of chemical shift overlap between farnesyl resonances, and example NOESY spectra for related atoms (e.g. methyls at positions 4, 10, and 14; other aliphatic protons at the other positions).

We would like to thank the reviewer for acknowledging the difficulty of assigning the farnesyl atoms, collecting unambiguous protein-farnesyl NOE restraints, and determining the farnesylated NMR structure. We therefore will provide the assignment strategy and more details in a manuscript draft that will be submitted soon. In addition, we provide the current draft of this manuscript as supplementary material for the reviewers.

Page 5: are protein-farnesyl NOEs observed for all specific interactions shown in Figure 2c? If yes, perhaps this should be stated (and if possible NOE connections plotted onto a structure for comparison, perhaps as a supplemental figure panel).

Figure 2c indeed displays experimental NOEs between PEX19 and the farnesyl group collected on PEX19 farnesylated samples using various labeling schemes on the protein side, as detailed in our answer on the previous comment. The figure legend has been corrected to state this and now reads: “*Two different views of the farnesyl recognition site. Dashed lines indicate intermolecular NOE correlations, collected on PEX19 farnesylated samples using various labeling schemes on the protein side.*”

Page 5: it is stated the NMR data show a distinct bent conformation of the isoprenoid. In the NMR structure calculations, were any specific restraints placed on the farnesyl group to ensure a specific conformation, for example? No detail is included in the CYANA structure calculations for how a farnesyl group was incorporated into the structure calculations.

We would like to thank again the reviewer for the clarifications requested. The farnesylated cysteine was parametrized by using the Dundee PRODRG2 Server. The CYANA library was prepared manually using the PDB coordinates derived from the PRODRG2 Server and all rotatable dihedral angles were defined. The dihedral angles for the isoprene double bonds were fixed to *trans* in accordance with our NMR analysis that defined the lipid as (2*E*,6*E*)-farnesol. For the last isoprenoid unit the methyls were stereospecifically assigned based on their carbon chemical shifts. Number 14 in *trans* the methyl with the upfield carbon frequency that is in the same frequency range as methyls 4 and 10 that are also in *trans* (~20 p.p.m.), and number 15 in *cis* the methyl with the downfield carbon frequency (~28 p.p.m.). For water refinement in CNS, the parameter and topology files were taken from the PRODRG2 Server and stereochemistry was the same as in CYANA. We stated above that a number of experiments were recorded to sample the unlabeled farnesyl moiety. However none of the farnesyl intramolecular NOESY spectra were used in structure calculations, in other words no farnesyl-farnesyl restraints were imposed during the structure determination. The farnesyl bent conformation is defined exclusively by the intermolecular NOE restraints from the protein and the allowed angular space of the farnesyl bonds. This information has been incorporated in the **Structure Calculation** section of the **Methods**.

Supplementary Figure 3: PDB codes should be listed for the structures used to compare to the farnesyl conformations. In particular, were any of these NMR-derived structures? If not, and if the compared structures were all derived from x-ray crystallography, could it be that the bent conformation could be an artifact of the NMR structure calculation? Are the farnesyl groups in the structures shown in this supplementary figure buried or solvent

exposed (it's difficult to tell from the figure)?

All structures displayed in Sup Fig. 3 are X-ray structures. In every structure the farnesyl group is buried in hydrophobic pockets formed by the proteins. In the X-ray structure of Aristolochene Synthase the buried farnesyl is also sharply bent confirming that the NMR conformation is not an artifact but a consequence of the protein pocket dimensions used by each protein to shield the farnesyl from the water. Figure and legend have been modified accordingly to state the PDB numbers and the method used (X-ray crystallography) to determine the structures shown

Page 5 and Figure 2e: despite the conformational rearrangement of the "lid" region, there is very little change in hetNOE and sPRE (Figure 1c). There could be some change in R1 or R2, but it is difficult to compare the relaxation parameters in Supplemental Figure 1c,d. Could a different plot be added to illustrate any dynamical differences in the lid region, or other regions in general, that occurs upon farnesylation?

In the crystal structure of the non-farnesylated PEX19 CTD (and likely in solution) the lid occupies the cavity which accommodates the farnesyl group after farnesylation. Thus, this region is not expected to be flexible or dynamic. This explains the little difference of heteronuclear NOE and sPRE data between farnesylated and non-farnesylated forms of PEX19.

Page 6: the text describing residues mutated to impair interactions the farnesyl moiety refer to Figure 2b, but I think it would be more appropriate to refer to Figure 2c.

We agree with the reviewer and now we refer to Figure 2c.

Page 6: how was it confirmed the PEX19 mutants were completely farnesylated in vitro?

Supplementary Figure 4a shows the NMR spectra of PEX19 mutants with or without farnesylation. For all mutants, upon farnesylation, no residual signals corresponding to non-farnesylated protein are visible. If any residual non-farnesylated PEX19 is still present, it is less than 5% of the total protein population.

Supplementary Figure 4a/b: it would be helpful if NMR spectral overlays were shown for each mutant with or without farnesylation.

NMR spectral overlays changed to show each mutant with or without farnesylation.

Supplementary Figure 4c/d: it would be helpful if wild-type PEX19 was shown for comparison. Also, were these plots generated for wild-type vs. mutant? This is not specified in the legend.

Chemical shift perturbations plotted in Supplementary Figure 4b (this was Supplementary Figure 4b in the original manuscript) are indeed for wild-type vs mutants for both non-farnesylated and farnesylated proteins. We clarified this in the figure legend accordingly.

Supplementary Figure 4f: it would be helpful if the difference volume listed under the chromatograms were labeled as such in the actual figure (or using a greek delta symbol). Otherwise this could be confused by the reader as an x-axis volume tick mark.

Thank you for this comment. This is corrected in the revised manuscript.

Figure 3a: how were the relative differences in hydrophobicity calculated? Is the unit of the y-axis a volume (mL)?

Comparison of farnesylated and unmodified proteins reveals a shift in the elution volume pointing to an increase of hydrophobicity due to farnesylation. As shown in Suppl. Fig. 4d for every protein (wild-type or mutants) a ΔV value is obtained from the elution maxima for the farnesylated and non-farnesylated protein. The relative differences in hydrophobicity of the mutants upon farnesylation were deduced by dividing the ΔV values with that of the wild-type. Therefore, the labeling of the y-axis refers to 'x-fold change in hydrophobicity', where wild type is set to 1.

Page 7: before discussing the functional complementation assay data (Figure 3b/c), it would be useful for the readers if a brief description is added after the second sentence about what this assay measures/detects and how this is related to PEX19 known function(s). Also, in Figure 3b/c the effect of wild-type vs. mutants in rescuing the (Δ)pex19 phenotype is shown, but I would think a control (empty plasmid/vector should also be shown to set the baseline that these proteins rescue the phenotype, similar to what is shown in Figure 5c.

We agree with the reviewer and added the following sentences: “We transfected PEX19-deficient cells with a bi-cistronic vector expressing PEX19 variants and eGFP-SKL as peroxisomal marker. The PEX19-deficient cells are characterized by the absence of import-competent peroxisomes, which is indicated by the mislocalization of the peroxisomal matrix marker to the cytosol and mislocalization of PEX14 to mitochondria. Complementation and thus reappearance of import-competent peroxisomes is indicated by an overlapping punctate pattern for eGFP-SKL and the peroxisomal membrane marker PEX14.”

The updated Figure 3b shows in the upper panel the control as requested by the reviewer.

Figure 4 and Supplementary Figure 5: while in general the data show that three different PMP peptides can bind the PEX19 CTD via NMR and microscale thermophoresis (MST), there are some inconsistencies between the experiments and addressing these would strengthen this section. For example, in Figure 4 the binding of a PMP peptide from ALDP is shown by NMR at a 5x and 10x molar equivalent added, but no MST data or analysis of NMR chemical shift perturbations is shown to confirm the binding affinity.

We have now confirmed the binding affinity for the ALDP peptide by using fluorescence polarization and found micromolar affinity for the farnesylated PEX19 CTD (new Suppl. Fig. 6) in support of our NMR titration experiments (Fig. 4b and 4c). The ALDP chemical shift perturbation analysis is shown in Fig. 4d for amide (Fig. 4b) and methyl groups (Fig. 4c) with respect to the PEX19 amino acid sequence. The NMR analysis allowed to identify critical residues for ALDP binding and design of mutants that disrupt PEX19 function.

In Supplementary Figure 5a, NMR data is shown for a PMP peptide from PEX13 at a 1.5x molar equivalent added, but in 5b the analysis is performed with a 0.5x molar equivalent added and again there is no analysis of binding affinity. Here, knowing the affinity would provide insight into the apparent intermediate exchange binding profile in 5a(right panel).

We could not determine the binding affinity for the PEX13 peptide due to severe line broadening occurring already at 0.5 molar ratio of the ligand. It was also not possible to calculate I/I_0 from the 1.5x ratio because all peaks disappear for farnesylated PEX19 CTD. Therefore, the analysis was performed at 0.5x ratio. Please note, that limited solubility of the PEX13 peptide and probable aggregation of the complex did also not allow alternative biochemical affinity measurement.

In contrast to the previous two PMP peptides, Supplementary Figure 5c shows MST binding affinity data a PMP peptide derived from PEX11b, but no NMR data are shown to map the interaction surface on the PEX19 CTD. MST data for PEX11b was selected because of their quality.

PEX11b NMR titrations were also performed and resulted in severe line broadening as PEX13 peptide. However, following the reviewer’s suggestion to provide K_D values for mutants and wildtype protein *in vitro* we used the fluorescence polarization assay for the ALDP derived peptide (Supplementary Figure 6), as for this PMP this was technically possible.

It is stated in the abstract (and the discussion) that farnesylation orchestrates the PMP binding region of PEX19, which includes helix $\alpha 1$, by an allosteric mechanism. However, while the farnesylation site is far removed from the PMP binding region in sequence space, it is nearby in 3D/structural space. Is this really a (long-range) allosteric mechanism, or is it merely that farnesylation directly stabilizes this region because of the nearby proximity to helix $\alpha 1$ -and therefore facilitates PMP peptide binding to the same surface? What does allosteric mean here?

It is true that both helix $\alpha 1$ and the lid are in close spatial proximity to the farnesyl group. Allostery refers to the fact that indeed the elements required for PMP binding are stabilized by the farnesyl group, which itself does not interact with the PMP.

Farnesylation is known (or at least suggested) to play a role in trafficking proteins to membrane surfaces, and PEX19’s ability to bind peroxisomal membrane proteins (PMPs) may suggest it could tether to a membrane. Here, farnesylation is shown to affect PMP interaction, but nothing is discussed about whether PEX19 can interact with membranes, and if it can, the possible role of the (buried) farnesyl group and the mechanism by which this it may become solvent accessible if that is required for membrane interaction?

We have added additional discussion of these points on p.11 in the revised manuscript: “We note that additional effects of PEX19 farnesylation could contribute to the functional activity of the Pex19 protein. 1) The presence of a farnesyl group may influence the PMP membrane insertion, i.e. support PMP cargo release by interactions of the farnesyl group with the membrane, PEX3 other factors. 2) It is also

important to note that farnesylation is only the first step in posttranslational modification of PEX19. It is conceivable that the farnesylated CAAX box is further processed by C-terminal proteolysis and carboxymethylation. This modification could be a way to regulate the activity of PEX19, as the carboxymethylation will further enhance the hydrophobicity of the C-terminal region and thus modulate interactions at the membrane. The presence of this PEX19 modification in cells has not yet been experimentally determined and its functional role should be investigated in future studies."

For the HADDOCK PEX19/ALDP PMP complex, it would be useful if the "HADDOCK SCORE vs. i-RMSD" figure plot is shown to complement the data provided in Supplemental Table S2.

We appreciate this suggestion. However, as the PMP interface essentially only involves recognition of two phenylalanine side chains the score vs. i-RMSD plot is not very instructive in this specific application. The two clusters reflect binding to the two different pockets, however, the binding surface exhibits quite significant variations as the docking of two phenylalanine side chains does not provide a continuous binding surface. Therefore, there are variations observed in the Score vs *i-RMSD* plot, which would give a misleading impression. Therefore, we prefer not to show this plot.

In all clusters the two pockets are "used" by phenylalanine side chains.

Just a reminder that PDB and BMRB accession codes should be added prior to publication when obtained.

PDB and BMRB codes have been obtained and listed in the manuscript.

Reviewer #1 (Remarks to the Author):

Nature Communications Ms. NCOMMS-16-08356A

Title: Allosteric Modulation of Peroxisomal Membrane Protein Recognition by Farnesylation of the Peroxisomal Import Receptor PEX19

Authors: Emmanouilidis, L. et al.

Remarks:

In the revised version, the authors provided additional data to reply to the concerns pointed out by the reviewer #1. New data with respect to the affinity to a PMP ligand *in vitro* and intracellular localization of PEX19 mutants supply deeper insight on the role of PEX19 C-terminal domain in the binding to PMP. However, this revised version does not contain actual data to explain the nearly undetectable difference of NMR signals in the helix $\alpha 1$ induced by PEX19 farnesylation (Fig. 1c). Furthermore, new data showing the defect of PEX19-M255R mutant in binding to a PMP ligand regardless of the farnesylation (new supplementary Fig. 6) raise another concern about the molecular property of M255R mutant as a reasonable model mutant. Therefore, it appears to the reviewer #1 that several points need to be addressed as listed below.

Reviewer's comments:

Major points:

Reply to the comment 2

In regard to the allosteric regulation of helix $\alpha 1$ by PEX19 farnesylation (Fig. 1c), the authors were unable to assign the NMR signals in helix $\alpha 1$ due to a signal overlap and line-broadening. Reviewer #3 also pointed out this issue, but no additional data was provided to explain nearly indistinguishable increase in heteronuclear NOE and no consistent decrease in sPRE in helix $\alpha 1$. Are there any alternative indications or methods available to support the allosteric conformational change of helix $\alpha 1$? The reviewer #1 believes that clear demonstration of the farnesylation-induced rearrangement of helix $\alpha 1$ is essential to convincingly assure the model that the authors claim in this paper.

Reply to the comment 3

While the addition of new supplementary Fig. 6 fairly clarifies the biochemical properties of the PEX19 mutants, it raises another significant concern, which needs to be addressed. The problem is that all three PEX19 CTD mutants, M179R, I195K, and M255R, showed significantly impaired binding to a PMP ligand as compared with the wild-type PEX19 CTD when these mutants were not farnesylated. This indicates that these mutations significantly affect the binding of PEX19 to the PMP ligand by mechanisms independent on the recognition of the farnesyl group. Thus, the observed *in vivo* effects of these mutations can be explained without assuming the allosteric effects by the recognition of the farnesyl group. Therefore, these mutations are inadequate for investigation of the contribution to the allosteric modulation of PMP recognition by PEX19 farnesylation *in vivo*. This reviewer suggests the authors to use other mutants that bind to the PMP ligand with an affinity comparable to that of the wild-type, non-farnesylated PEX19.

Reply to the comment 4

The authors describe possible mechanisms to rationalize the apparently small differences in the heteronuclear NOE data between the wild-type and M255R mutant PEX19 CTDs with farnesylation. Although there are probably the truth to some extent in what they address, the explanation still remains somewhat ad hoc. The authors also pointed out the difficulty in dealing with the M255R

mutant. This again such difficulty likely reflects an aspect that the mutation significantly alters the property of the PEX19 CTD rather than its farnesyl-group recognition property as this reviewer points out in the previous comment in the author's reply to the comment 3. Thus, this reviewer suggests the authors to use another mutant that behaves like the wild-type protein with no effects on its farnesyl-group recognition property and supports the authors' conclusion. Since the current conclusion is based on a relatively small difference of a single terminal residue in the heteronuclear NOE data between the wild-type PEX19 CTD and only one mutant, the additive and suitable information obtained from such distinct mutant is required and highly beneficial.

Reply to the minor comment 4

This reviewer meant that the authors needed to indicate the concentrations of PEX19 CTD and PMP ligands used in the experiments and better to describe in the legend for Figs. 4a and 4b, not Supplementary Figs. 4a and 4b.

Additional minor comment

For non-specialists and general readers, it would be better to clearly describe that eGFP-SKL and eGFP-PTS1 are the same construct.

REVIEWERS' COMMENTS:

Reviewer #2 (Remarks to the Author):

The authors have satisfactorily addressed all my previous concerns. Please correct the following (minor) textual errors:

Lines 252-254: "The amide and methyl signals affected by the titration cluster to the hydrophobic groove that is formed on the PEX19 surface only upon farnesylation (Fig. 2b; Fig. 4d,e)." This sentence does not read well.

Line 513: replace "13.000 rpm" by "13,000 rpm".

Legend to Supplementary Fig. 5: replace "10 mm" by "10 μm ".

Reviewer #3 (Remarks to the Author):

The authors have addressed all of my concerns in this revised manuscript, and I support publication of this manuscript in Nature Communications.

Point-by-point response

Reviewer #1

In the revised version, the authors provided additional data to reply to the concerns pointed out by the reviewer #1. New data with respect to the affinity to a PMP ligand *in vitro* and intracellular localization of PEX19 mutants supply deeper insight on the role of PEX19 C-terminal domain in the binding to PMP. However, this revised version does not contain actual data to explain the nearly undetectable difference of NMR signals in the helix $\alpha 1$ induced by PEX19 farnesylation (Fig. 1c). Furthermore, new data showing the defect of PEX19-M255R mutant in binding to a PMP ligand regardless of the farnesylation (new supplementary Fig. 6) raise another concern about the molecular property of M255R mutant as a reasonable model mutant. Therefore, it appears to the reviewer #1 that several points need to be addressed as listed below.

Reviewer's comments:

Major points:

Reply to the comment 2: In regard to the allosteric regulation of helix $\alpha 1$ by PEX19 farnesylation (Fig. 1c), the authors were unable to assign the NMR signals in helix $\alpha 1$ due to a signal overlap and line-broadening. Reviewer #3 also pointed out this issue, but no additional data was provided to explain nearly indistinguishable increase in heteronuclear NOE and no consistent decrease in sPRE in helix $\alpha 1$. Are there any alternative indications or methods available to support the allosteric conformational change of helix $\alpha 1$? The reviewer #1 believes that clear demonstration of the farnesylation-induced rearrangement of helix $\alpha 1$ is essential to convincingly assure the model that the authors claim in this paper.

As we stated in our previous response to reviewer #3, helix $\alpha 1$ is present in solution in both forms of PEX19, farnesylated and non-farnesylated. Heteronuclear NOEs report on local flexibility of the amides involved and are not expected to differ for the amides in helix $\alpha 1$ in the farnesylated and non-farnesylated form, as the helix is (at least partially) formed in either case, despite the different orientation. Thus, the similar heteronuclear NOE values are consistent with our statement that this helix already exists in the non-farnesylated protein. The sPRE data, which refer to solvent accessibility of amides, are increased for residues in helix $\alpha 1$ in the non-farnesylated protein but are lower compared to the fully exposed C-terminal tail. This indicates that helix $\alpha 1$ is not completely solvent exposed in absence of farnesylation, and might weakly interact with the globular core domain of the Pex19 CTD. For two amides in helix $\alpha 1$, which could be unambiguously analyzed, additional protection upon farnesylation is clearly observed. This is consistent with a rigidification and packing of helix $\alpha 1$ against the core domain. To clarify the analysis and interpretation of these data the sentence referring to Fig 1c has been changed to: "These data demonstrate that farnesylation induces substantial rigidification and compaction of the PEX19 CTD that primarily stabilizes the C-terminal region but also locks the arrangement of helix $\alpha 1$ with respect to the core domain."

A clear demonstration of the farnesylation-induced rearrangement of helix $\alpha 1$ is, however, provided by the high-resolution structure that we present in the manuscript. To unambiguously probe the PEX19-farnesyl contacts we prepared samples with different labeling schemes that allowed to collect a large number of protein-farnesyl NOEs, involving residues in helix $\alpha 1$. The overall fold, including the specific orientation of helix $\alpha 1$, is also determined by NOE contacts that do not involve the farnesyl group. These NOEs are incompatible with the crystal structure of the non-farnesylated protein. In the non-farnesylated structure residues of helix $\alpha 1$ (M179-L183) pack against L252 of helix $\alpha 4$. In the farnesylated protein helix $\alpha 1$ is further away from helix $\alpha 4$ (due to a farnesylation-induced rearrangement of helix $\alpha 1$) because the space in between is occupied by the bulky residues following the farnesylated cysteine (C296far-L297-I298). In particular L297 is in contact with residues in helix $\alpha 1$ (L183-M179) and I298 in contact with residues in helix $\alpha 4$ (L252-Q256). Therefore, the conformation of helix $\alpha 1$ in the farnesylated protein is clearly distinct from the crystal structure of the non-farnesylated protein.

Our NMR data indicate that in the non-farnesylated protein the orientation of helix $\alpha 1$ in solution is also different from the crystal structure and, in fact, somewhat flexible. Farnesylation then locks helix $\alpha 1$ in a specific conformation and thereby forms a high-affinity binding site for PMPs. In our view, this is clear evidence of an allosteric modulation of the PMP binding site, which is driven by farnesylation.

Reply to the comment 3: While the addition of new supplementary Fig. 6 fairly clarifies the biochemical properties of the PEX19 mutants, it raises another significant concern, which needs to be addressed. The problem is that all three PEX19 CTD mutants, M179R, I195K, and M255R, showed significantly impaired binding to a PMP ligand as compared with the wild-type PEX19 CTD when these mutants were not farnesylated. This indicates that these mutations significantly affect the binding of PEX19 to the PMP ligand by mechanisms independent on the recognition of the farnesyl group. Thus, the observed *in vivo* effects of these mutations can be explained without assuming the allosteric effects by the recognition of the farnesyl group. Therefore, these mutations are inadequate for investigation of the contribution to the allosteric modulation of PMP recognition by PEX19 farnesylaton *in vivo*. This reviewer suggests the authors to use other mutants that bind to the PMP ligand with an affinity comparable to that of the wild-type, non-farnesylated PEX19.

The binding experiments presented in Supp. Fig. 6 corroborate previous observations, stating that PEX19 farnesylation *enhances* binding to PMP cargo. For the peptide we used in our *in vitro* studies we obtained a seven-fold increase in affinity for the wild-type protein upon farnesylation. The fact that non-farnesylated PEX19 is able to bind PMP ligands, albeit with lower affinity, indicates that a compromised binding surface is present in the absence of farnesylation. Helix $\alpha 1$ is directly involved in PMP interactions, thus mutations in this helix are expected to reduce the binding in both the non-farnesylated protein (where the helix does not adopt an optimal conformation for binding) and in the farnesylated protein (where the helix is locked in a conformation that is optimized for PMP interactions). The crystal structure of non-farnesylated PEX19 and our analysis shows that the protein conformation differs with or without farnesylation. Our high-quality structure explains the enhancement in PMP binding by an allosteric mechanism that locks helix $\alpha 1$ (and the lid segment) to a defined position that is profoundly different compared to the crystal structure (even though the orientation of helix $\alpha 1$ in the crystal likely is enforced by crystal packing), but is not populated in solution in the absence of farnesylation. In any case, farnesylation reshapes the surface around helix $\alpha 1$ where binding takes place. The M179R and I195K mutants indeed confirm this because alter the properties of the binding surface (hydrophobicity) in absence or presence of farnesylation. M255 is distant from the binding surface. From the HSQC chemical shift perturbations it can be seen that M255R substitution has large effects on the protein conformation in both forms, non-farnesylated and farnesylated and compromises PMP binding for both. Similar structural effects, but to smaller extent, are also seen for M179R and I195K mutants. The reason is that the hydrophobic residues that recognize farnesyl, in its absence are engaged in mutual hydrophobic interactions to shield themselves from the solvent (at least as seen in the crystal structure). The importance of M255 can only be deduced with the knowledge of the farnesylated structure.

Our data suggest that it should not be possible to design a mutation that affects PMP binding of the farnesylated protein but does not impair PMP binding of the non-farnesylated form, as the conformation of helix $\alpha 1$, stabilization of the farnesyl group and the PMP binding site are tightly interconnected. The fact that mutations in the helix in both farnesylated and non-farnesylated Pex19 show reduced PMP interactions, however, at a different absolute level, is fully consistent with an “enhancing” contribution linked to farnesylation.

Reply to the comment 4: The authors describe possible mechanisms to rationalize the apparently small differences in the heteronuclear NOE data between the wild-type and M255R mutant PEX19 CTDs with farnesylation. Although there are probably the truth to some extent in what they address, the explanation still remains somewhat ad hoc. The authors also pointed out the difficulty in dealing with the M255R mutant. This again such difficulty likely reflects an aspect that the mutation significantly alters the property of the PEX19 CTD rather than its farnesyl-group recognition property as this reviewer points out in the previous comment in the author’s reply to the comment 3. Thus, this reviewer suggests the authors to use another mutant that behaves like the wild-type protein with no effects on its farnesyl-group recognition property and supports the authors’ conclusion. Since the current conclusion is based on a relatively small difference of a single terminal residue in the heteronuclear NOE data between the wild-type PEX19 CTD and only one mutant, the additive and suitable information obtained from such distinct mutant is required and highly beneficial.

For all mutants tested, including M255R, the NMR chemical shift perturbations indicate that the protein fold is not affected by any of the mutations (Suppl. Fig. 4). The hydrophobicity analysis shows that none buries the farnesyl as the wild-type does, indicating local changes, that are also seen by chemical shift differences in the NMR spectra. Designing a mutation that does not affect farnesyl recognition but compromises PMP binding is difficult, as our structure shows that the PMP binding surface is linked with the farnesyl binding cavity. However, we suggest that the P273F/P274F mutant represents a mutation that affects largely solvent exposed residues in the lid and have no direct contacts with the farnesyl, but -

as shown by our *in vivo* experiments – nevertheless impair peroxisomal biogenesis.

Reply to the minor comment 4: This reviewer meant that the authors needed to indicate the concentrations of PEX19 CTD and PMP ligands used in the experiments and better to describe in the legend for Figs. 4a and 4b, not Supplementary Figs. 4a and 4b.

Concentrations are now indicated in figure legend.

Additional minor comment: For non-specialists and general readers, it would be better to clearly describe that eGFP-SKL and eGFP-PTS1 are the same construct.

Thank you for pointing this out. This has been clarified in the text on page 7.

Reviewer #2 (Remarks to the Author):

The authors have satisfactorily addressed all my previous concerns. Please correct the following (minor) textual errors:

Lines 252-254: "The amide and methyl signals affected by the titration cluster to the hydrophobic groove that is formed on the PEX19 surface only upon farnesylation (Fig. 2b; Fig. 4d,e)." This sentence does not read well.

The sentences was corrected and now reads : The amide and methyl signals affected by the titration cluster to the hydrophobic groove that is formed on the farnesylated PEX19 surface (Fig. 2b; Fig. 4d,e)

Line 513: replace "13.000 rpm" by "13,000 rpm".

Corrected

Legend to Supplementary Fig. 5: replace "10 mm" by "10 μ m".

Corrected

Reviewer #3 (Remarks to the Author):

The authors have addressed all of my concerns in this revised manuscript, and I support publication of this manuscript in Nature Communications.